# Impacts of freshwater changes on Antarctic sea ice in an eddy-permitting sea ice-ocean model

Verena Haid[1,2], Doroteaciro Iovino[1] and Simona Masina[1,3]

[1]Fondazione Centro Euro-Mediterraneo sui Cambiamenti Climatici (CMCC), Bologna, Italy
[2]Now at: Laboratoire de Oceanographie Physique et Spatiale (LOPS), CNRS, Plouzané, France
[3]Istituto Nazionale di Geofisica e Vulcanologia (INGV), Bologna, Italy

*Correspondence to*: Verena Haid (v.haid@web.de)

**Abstract.** In a warming climate, satellite data indicate that the sea ice extent around Antarctica has increased over the last decades. One of the suggested explanations is the stabilizing effect of increased mass loss of the Antarctic ice sheet. Here, we
investigate the sea ice response to changes in both the amount and the spatial distribution of freshwater input to the ocean by comparing a set of numerical sensitivity simulations with additional supply of water at the Antarctic ocean surface. We analyse the short-term response of the sea ice cover and the on-shelf water column to variations in the amount and distribution of the prescribed surface freshwater flux.

Our results confirm that enhancing the freshwater input can increase the sea ice extent. Only for "extreme" freshwater
additions, our experiments show a negative development of the sea ice extent. We find that the spatial distribution of the fresh water is of great influence on sea ice concentration and thickness as it affects sea ice dynamics and thermodynamics. For strong regional contrasts in the freshwater addition the dynamic response dominates the local change in sea ice, which generally opposes the thermodynamic response. Furthermore, we find that additional coastal runoff generally leads to fresher and warmer dense shelf waters.

## 1 Introduction

Sea ice is one of the key components of the polar climate system controlling air-ocean exchange and driving deep ocean convection. Even though satellite observations show a prevailing warming of the surface air (Jones et al., 2016) and the upper-ocean (Gille 2008), they also indicate that Antarctic sea ice has been expanding over the past few decades (e.g. Vaughan et al., 2013). The positive net (circumpolar-averaged) trend is the sum of partly opposing regional trends of the
same order of magnitude (Parkinson and Cavalieri, 2012). The causes of this positive trend in a generally warming world are still debated. Several mechanisms have been proposed to explain the expanding Antarctic sea ice. Many studies, for example, attribute the increase of sea ice to changes in the circumpolar wind field. The strengthening of the circumpolar westerly winds alters the sea ice drift patterns and could result in the regionally different trends observed in the sea ice cover (e. g. Thompson and Solomon, 2002; Liu et al., 2004; Lefebvre and Goosse, 2005; Turner et al., 2009). An increase of
precipitation over the Southern Ocean has influence on surface salinity, albedo of ice covered areas and ice thickness by

submersion and could also be a possible contributor to the observed increase in Antarctic sea ice extent (Liu and Curry, 2010).

Zhang (2007) suggests that the trends in the Antarctic sea ice extent can be explained as the result of a feedback between the sea ice and the upper ocean stratification. He attributes the increase in sea ice to a reduced capability of the ocean to melt sea ice. Enhanced surface air temperatures and longwave radiation in his model lead to increased stratification and reduced vertical heat flux in the ocean. Similarly, Goosse and Zunz (2014) argue that the multidecadal variability of the Southern Ocean can explain the recent trends in Antarctic sea ice in connection with small changes of the pycnocline that increase the ocean's stratification. Also Bintanja et al. (2013) attributes the change in sea ice to a fresher surface layer, but sees the cause in the enhanced influx of meltwater from the Antarctic ice sheet.

The mass loss of Antarctic ice sheets by basal melt has recently been found to be accelerating (Jacobs et al., 2011; Pritchard et al., 2012) leading to the freshening of the mixed layer and thus to a stronger stratification of the water column. This shields the surface more effectively from the heat stored in the deeper layers of the ocean and therefore sea ice melt is reduced and sea ice growth is furthered. The importance of adding glacial melt water from the Antarctic continent to the Southern Ocean in simulations to account for ice-shelf melt water has been indicated by e.g. Hellmer (2004) and Stössel et al. (2007). However, so far, only few studies have been conducted to investigate the effect that changing Antarctic melt water provokes in the Southern Ocean sea ice.

Bintanja et al. (2013, 2015) investigated the sensitivity of Antarctic sea ice to an increase of freshwater flux intended to re-produce current sources from Antarctic ice shelf melt. Bintanja et al. (2013) suggest that including realistic changes in the Antarctic ice sheet mass in a coupled climate model can lead to a better simulation of the evolution of the sea ice in the Southern Ocean. In their study, with a freshwater increase of 250 Gt yr$^{-1}$ under constant year 2000 forcing, sea ice concentration increased up to 10 % over a ~30year period, suggesting that the net sea ice trend is dominated by the increased ice-shelf melt, while winds may be responsible for the regional trends. Bintanja et al. (2015) then assessed the effect of increased (basal) melt rate of the Antarctic ice sheet and the associated freshwater flux on future sea-ice trends in a RCP8.5 scenario. In their coupled simulations, projected mass losses of the Antarctic ice sheet impact the future sea-ice trend: an additional freshwater forcing (120 Gt yr$^{-1}$) is necessary to reverse the sign of the sea ice trend to positive values.

On the other hand, Swart and Fyfe (2013) showed that an accelerating rate of freshwater forcing, with a magnitude constrained by observations, is unlikely to impact significantly the Antarctic sea ice trends simulated by the CMIP5 models, and that the freshwater addition is unlikely to reproduce the spatial pattern of the observed trends. More recently, Zunz and Goosse (2015) investigated the dependence of the forecasting skill of an Earth-system model on the freshwater input. Their results show a strong dependency on the initial state and, in consequence, suggest that neither atmospheric nor freshwater trends cause the current sea ice trends, but that the ocean's preconditioning of the 1970s lead to surface cooling and sea ice expansion.

Merino et al. (2016) used an iceberg model coupled to a sea ice-ocean model to establish a seasonal climatology of iceberg melt for the Southern Ocean. They find that the iceberg melt water leads to higher sea ice concentration and thickness, with

exception of the Amundsen/Bellingshausen Sea area. Pauling et al. (2016) employed an Earth-system model to investigate the effect of artificially augmented, constant freshwater input on sea ice. They tested the sensitivity to freshwater additions from current estimates to much larger values (3000 Gt yr$^{-1}$) and compared an iceberg model-based surface distribution with a coastal distribution at the depth of the ice shelf front. Their model experiments show that enhancing the freshwater input by an amount within the range of estimates of the Antarctic mass imbalance does not significantly affect sea ice area, magnitude or trend. Further, their results show the sea ice response to be insensitive to the depth of freshwater injection.

Our study investigates the short-term response of sea ice and upper-ocean to prescribed changes of the surface freshwater input, both in magnitude and spatial distribution. We do consider these modifications as a perturbation of the current-climate environment. We specifically include the dynamic response of the sea ice-ocean system in our analysis. Generally, the previous studies use very crude renderings of the spatial distribution of the freshwater addition. Here, we focus on the differences between a widely-used uniform runoff distribution around Antarctica and more complex spatially varying distributions. In our study, we employ an eddy-permitting ocean-sea ice model. Four experiments with differing spatial distribution and magnitude of the Antarctic surface freshwater flux are analysed with respect to the response of sea ice concentration, thickness, and velocity in space, and extent, volume and production over time. In addition, the development of the on-shelf water column and the dense shelf water at the main sites of dense shelf water formation are presented. While these changes do not directly correspond to the trends observed around Antarctica in recent decades, they tell us how the sea ice reacts dynamically and thermodynamically to spatially limited changes in the freshwater input. Thus, they provide a measure of what to expect as a sea ice response to observed changes in the runoff, and also offer explanations for observed changes in sea ice and water properties. Three additional experiments with enhanced freshwater amounts exceeding the range of current estimates (70-290 Gt yr$^{-1}$; Rignot et al., 2008; Joughin and Alley, 2011; Shepherd et al., 2012; Vaughan et al., 2013; Wouters et al., 2013; Rignot et al., 2013; Velicogna et al., 2014), allow us to gain insight into possible future sea ice changes.

## 2 Methods

### 2.1 Model description

The presented numerical calculations are based on version 3.4 of NEMO (Nucleus for European Modelling of the Ocean) global general circulation model (Madec et al., 2012). The ocean component is a finite difference, hydrostatic, primitive equation ocean general circulation model. Our configuration employs a global ORCA025 tripolar grid (Madec and Imbard, 1996) whose horizontal resolution is 0.25° (approximately 27.75 km) at the Equator and increases with latitude to be e.g. ≈10 km at 70°S (1442 grid points × 1021 grid points). The vertical grid has 75 levels, the spacing of which increases with a double tanh function of depth from 1m near the surface to 205m at the bottom, with partial steps representing the bottom topography (Barnier et al., 2006).

The model bathymetry is based on the combination of ETOPO1 data set (Amante and Eakins, 2009) in the open ocean and GEBCO (IOC, IHO and BODC, 2003) in coastal regions. Hand editing is performed in a few key areas.

The model uses a linear free surface and an energy and enstrophy conserving momentum advection scheme. The horizontal viscosity is bi-Laplacian with a value of $1.8 \times 10^{11}$ m$^4$ s$^{-1}$ at the Equator, reducing poleward as the cube of the maximum grid cell size. Tracer advection uses a total variance dissipation (TVD) scheme (Zalesak, 1979). Laplacian lateral tracer mixing is along isoneutral surfaces with a coefficient of 300 m$^2$ s$^{-1}$. The vertical mixing of tracers and momentum is parameterized using the turbulent kinetic energy (TKE) scheme. Subgrid-scale vertical mixing processes are represented by a background vertical eddy diffusivity of $1 \times 10^{-5}$ m$^2$ s$^{-1}$ and a globally constant background viscosity of $1 \times 10^{-4}$ m$^2$ s$^{-1}$. The bottom friction is quadratic. A diffusive bottom boundary layer scheme is included.

The sea ice component is the Louvain-la-Neuve Sea Ice Model, LIM2 (Fichefet and Morales Maqueda, 1997), which includes the representation of both the thermodynamic and dynamic processes. It accounts for sensible heat storage within the ice. The vertical heat conduction is calculated assuming two layers of ice and a snow layer on top. Sub-grid scale thickness distributions are thereby accounted for by use of an effective conductivity. The model also includes the conversion of snow to ice if the ice surface is depressed under the sea surface by the snow load. The ice dynamics are calculated according to external forcing from wind stress, ocean stress and sea surface tilt and internal ice stresses using C grid formulation (Bouillon et al., 2009). The elastic-viscous-plastic (EVP) formulation of ice dynamics by Hunke and Dukowicz (1997) is used.

The model is forced with ERA-Interim global atmospheric reanalysis (Dee et al., 2011), with 0.75°×0.75° spatial resolution. The turbulent variables are given as 3-hour mean values, while the radiative fluxes and precipitation are given as daily mean. The surface boundary conditions are prescribed to the model using the CORE bulk formulation proposed by Large and Yeager (2004). The forcing routine and the ice model are called every 5 time steps of the ocean model (every 90 minutes).

A suite of simulations is presented in this manuscript: a control run (hereafter CTR) and seven sensitivity experiments (S1-S4, S2-Incr, S2-High and S3-High). CTR was started from a state of rest in January 1979 and run for 35 years. Initial conditions for temperature and salinity are derived from the World Ocean Atlas 2013 climatological fields (Zweng et al., 2013; Locarnini et al., 2013), merged with PHC2.1 climatology over the Arctic region. The initial condition for the sea ice was inferred from the NSIDC Bootstrap products for January 1989. All freshwater experiments are branched off from CTR in January 2004 and run for ten years. In these simulations, we changed the amount and/or distribution of the Antarctic runoff; all other settings are identical to CTR.

The runoff data is a monthly climatology derived from the Global River Flow and Continental Discharge Data Set (Dai and Trenberth, 2002; Dai et al., 2009) for the major rivers and estimates by Jacobs et al. (1992) for the Antarctic coastal freshwater discharge. It has been adapted to the ORCA025 grid and applied along the land mask (Bourdalle-Badie and Treguier, 2006). It includes 109 major rivers and a coastal runoff with a global mean value of 1.26 Sv. The fresh water is added at the surface with zero salinity and at sea surface temperature. In the areas of freshwater addition, the vertical diffusion is enhanced (mixing coefficient: $2 \times 10^{-3}$ m$^2$ s$^{-1}$) over a depth of 15 m. In the Southern Ocean, we vary the runoff

field between simulations, as described in the next section. The runoff follows a seasonal cycle, which is unaltered relative to the mean amount in all experiments (Figure 1a). No surface restoring for tracers was used in the simulations. The simulation was run without any constraint on the freshwater budget.

## 2.2 Experiment design

To test the response of sea ice to changes in the melting of glacial ice around Antarctica, we present seven sensitivity experiments in this study, where the surface freshwater input is modified in its magnitude and spatial distribution. It is worth mentioning that our study does not attempt to closely reproduce reality, but aims to investigate the response of our ocean-sea ice system to an additional forcing. Ice shelves and icebergs are not explicitly resolved in our configuration; therefore any source of melt water is represented in the runoff field. The sensitivity experiments are run from 2004 to 2013. The 10-year

period is not sufficient to let the 3D ocean reach equilibrium after the disturbance at the surface. However, this study only aims to examine the very-short-term response of the sea ice and upper ocean system to the imposed idealized changes in the Southern Ocean freshening, via the comparison to the CTR run. A short overview of the experiments and their differences is also given in Table 1.

The CTR total runoff represents a continental discharge of 2610 Gt yr$^{-1}$, and is uniformly distributed along the Antarctic

coastline (Figure 1b), as commonly done in ocean models. The value is close to observation-based estimates of 2760 Gt yr$^{-1}$ by Rignot et al. (2013), 2775 Gt yr$^{-1}$ by Depoorter et al. (2013), and 2260 Gt yr$^{-1}$ by Liu et al. (2015), that include both basal melt and iceberg contributions. For information on the general performance of CTR please refer to the supplementary material.

In the first sensitivity experiment, S1, the magnitude of freshwater input is increased by 5 % adding 130 Gt yr$^{-1}$, and is

spatially constant as in CTR (Figure 1b). A comparison between the simulations allows us to study the effect of increased runoff without interference of other factors. The amount of increase is a conservative choice within the range of recent estimations of Antarctic mass loss (e. g. Shepherd et al., 2012; Vaughan et al., 2013; Wouters et al., 2013; Velicogna et al., 2014).

S2 simulation introduces a more realistic uneven spatial distribution of the runoff based on estimates of basal melt and

calving by Rignot et al. (2013). The runoff, still distributed close to the coastline, varies in magnitude by region (Figure 1c). In some areas (mainly East Antarctica), it is reduced compared to CTR, while in other areas (e.g. Weddell Sea and Amundsen Sea) it is strongly increased. The total freshwater flux is increased by 150 Gt yr$^{-1}$ compared to CTR.

S3 takes into account that fresh water does not enter the ocean exclusively at the coastline. Meltwater input from icebergs is introduced offshore over a much wider area. Only a reduced amount of runoff (1670 Gt yr$^{-1}$) is distributed close to the

Antarctic coastline to represent ice shelf melt (following the S2 distribution), while 1090 Gt yr$^{-1}$ is associated to the iceberg freshwater release (Rignot et al., 2013), and is widely distributed in the Southern Ocean (with four levels of flux intensity) (Figure 1d). The shape of this distribution is loosely based on iceberg drift and melt studies, e.g. Gladstone et al. (2001), Silva et al. (2006) and Jongma et al. (2008). The total amount of runoff is 2760 Gt yr$^{-1}$ as in S2.

S4 features a more extreme distribution of runoff that focuses on the key areas of dense water formation. Since the sea ice formation processes over the Antarctic continental shelves are essential factors in the formation of dense shelf water and consequently of the bottom water of the world ocean, the effect of runoff on the water column in these areas is of special interest. The S4 runoff adds 420 Gt yr$^{-1}$ to the CTR runoff, but distributes it all in only three locations: in front of the Filchner/Ronne Ice Shelf in the Weddell Sea (230 Gt yr$^{-1}$), in front of the Ross Ice Shelf in the Ross Sea (120 Gt yr$^{-1}$) and in front of the Amery Ice Shelf in Prydz Bay (60 Gt yr$^{-1}$) (Figure 1e). The total of the runoff is 3030 Gt yr$^{-1}$.

We present three additional experiments in which the amount of additional fresh water exceeds the range of current Antarctic mass loss estimates. These runs are an attempt to determine how much fresh water is required to have a significant effect on the sea ice area trend, in our model. The spatial runoff distribution in the experiments S2-Incr and S2-High is based on S2 simulation, while the experiment S3-High follows the S3 distribution including the widespread offshore freshwater addition. In S2-Incr, the runoff increases from 2760 Gt yr$^{-1}$ in 2004 to 3310 Gt yr$^{-1}$ in 2013 in 4 steps (137 Gt yr$^{-1}$ every 2 years). In S2-High and S3-High, a constant freshwater input of 3310 Gt yr$^{-1}$ is added around Antarctica in different spatial distributions.

## 3 Sea ice response to freshwater modifications

In this section, the impact of modifications in the freshwater supply on the sea ice is analyzed by comparing the sensitivity experiments S1-S4 with the reference simulation CTR. Overall, CTR properly reproduces the mean state of upper-layer ocean and sea ice. The seasonal cycle and spatial distribution of sea ice concentration compare well with satellite observations. Temperature and salinity are close to observations from the World Ocean Circulation Experiment (Orsi and Whitworth, 2005; see supplementary material for a more detailed description of CTR results).

In the comparison, we focus our analysis on the 6-month period from April to September when the sea ice properites in CTR are closest to observations. In the following, the word winter referring to a specific time period identifies the period April – September.

In Sect. 3.1, we analyse the response of sea ice concentration, thickness and velocity to artificial freshwater enhancement in the experiments S1-S4; while in Sect. 3.2 we discuss the time series of ice extent, volume and ice production.

### 3.1 Spatial response patterns

Since sea ice concentration reaches its maximum during the winter months (Figure 2a or CTR), there is only limited leeway for it to increase under additional freshwater input in the sensitivity runs. The maximum changes are found in the marginal ice zone (Figure 2, left column). The changeability in time of the ice concentration differences between experiments S1-S4 and CTR is comparatively high, so areas with a high statistical confidence level are limited. Changes of the sea ice thickness (Figure 2, middle column) generally exhibit a similar, but spatially more coherent pattern compared to the changes in sea ice concentration: areas of higher (lower) concentration yield thicker (thinner) sea ice. The longer-lasting character of the

changes in thickness reduces also the changeability of the ice thickness differences in time and increases the areas of statistical significance compared to the changes in concentration.

Sea ice velocity is also affected by the changes in runoff. The addition of fresh water locally increases the sea surface height (SSH) in two ways: 1) directly by the addition of the fresh water's volume to the ocean and 2) by decreasing the density of the upper water column. This affects both the SSH slope and the density gradient, which influences the surface current of the ocean and the sea ice drift. Changes in the sea ice velocity influence the sea ice thickness. By an increased drift velocity, the time available for thermodynamic growth can be shortened, but in zones of sea ice convergence the thickness is increased through the dynamic compaction processes such as rifting and rafting. Therefore, changes in the amount of additional fresh water directly affect sea ice properties by both thermodynamic and dynamic processes. The changes in sea ice velocity caused by the freshwater input modify the ice thickness due to the advection speed and dynamic compaction. The areas featuring a high statistical confidence level for the changes in sea ice velocity (Figure 2, right column) are predominantly found along the coastline, where the runoff addition per area is higher and the coastal current distributes it primarily within a narrow band around the continent. Most of the offshore velocities show seemingly erratic changes induced by the highly variable fronts and eddies in the Antarctic Circumpolar Current.

### 3.1.1 S1: Response to a simple runoff increase

In the S1 experiment, the sea ice concentration and thickness feature small changes from CTR. The former shows a more relevant increase at the tip of the Antarctic Peninsula (Figure 2d). The latter thickens mainly in the western Ross Sea, west of and at the tip of the Antarctic Peninsula, and in the central Weddell Sea (Figure 2e), while larger areas of sea ice thinning are in the eastern Ross Sea and the southwestern Weddell Sea. The surface freshening by the enhanced runoff reduces the density of the surface layer and therefore leads to a strengthening of the ocean stratification (not shown), which inhibits vertical heat transport from the deeper ocean to the surface. The SST cools and the freezing temperature increases, both of which are expected to increase the ice formation and cover. However, we have to consider the ice dynamics in order to explain local maxima and especially decreases in either concentration or thickness. In S1, the freshwater increase along the coastline strengthens the coastal current and the coastal sea ice drift is slightly sped up compared to CTR (Figure 2f). The faster ice drift leaves some areas with younger and thus thinner ice. In areas with a more complex coastline geometry, it causes stronger convergence and compaction of the ice, thus resulting in higher ice concentrations and thicker ice.

### 3.1.2 S2: Response to strong regional runoff variations

The runoff distribution used in S2 introduces regionally-varied coastal surface freshwater fluxes. The responses of the sea ice properties can therefore be expected to be strongly region-dependent. For the sea ice concentration (Figure 2g), we find changes of high statistical confidence in the coastal area. Increases in ice concentration and thickness (Figure 2h) occur in the eastern Weddell Sea, in the western Ross Sea, close to the coast of East Antarctica, and east of the tip of the Antarctic Peninsula. Areas of strongly reduced sea ice are located adjacent to the coast of the Amundsen and eastern Ross Seas and in

the southern Weddell Sea. Since in S2, the freshwater input is varied regionally along the coastline also the ice drift velocities are altered dependent on the location (Figure 2i). Compared to CTR, the westward ice drift is faster along the coast of the Amundsen and Ross seas. From the Prydz Bay to the southern Weddell Sea it is slower than in CTR. From here, sea ice speeds up compared to CTR, moving northward along the Antarctic Peninsula, to slow down again on the western side of the peninsula toward the Bellingshausen Sea. In S2, the changes in sea ice velocity cause most of the local changes in sea ice concentration and thickness.

To investigate more closely the mechanisms controlling regional sea ice behaviour in S2, we subdivided the widely-used 5 sectors of the Southern Ocean (e.g. Parkinson and Cavalieri, 2008, 2012) into 10 regions (Figure 1b). With the exception of the western and eastern Weddell regions, which both have a width of 40° in longitude, all regions span 35°. A northern limit was also employed, chosen individually for every region, in a way to include areas under the influence of the westward coastal current, while excluding most of the areas with eastward sea ice drift.

The compilation of the regional differences in runoff and sea ice characteristics between S2 and CTR (Table 2) confirms that the regional thermodynamic response to an increase (decrease) of runoff is an increase (decrease) of sea ice production. Only the western Ross Sea region (WRoS) is an exception to this rule, because here the increase of the runoff along the southern coastline is exceeded by the reduction of runoff along the north-south directed western coastline (Figure 1c). For the sea ice production, however, the southern coastline is of greater influence because of frequent polynya activity due to southerly winds.

The change in sea ice presence (concentration, thickness and volume) in most regions is contrary to the thermodynamic response. This strongly suggests that the impact on sea ice presence in S2 is locally determined by the response of the sea ice dynamics, and that regional thermodynamics play a minor role. The differences in the sea ice velocities impact the regional rates of sea ice import and export. Therefore, given strong regional contrasts of the freshwater addition, the dynamic response decides the development of sea ice presence in the area. With the S2 freshwater input, two regions show a distinct behaviour: the Wilkes Land (WiL) and the Bellingshausen Sea (BeS) sectors (Table 2). Both feature an increase in runoff, sea ice production, sea ice concentration and thickness. In WiL, the coastal current is dominated by the larger scale situation; although in the Amundsen and Ross Seas the current increasingly gains speed compared to CTR, while circling East Antarctica the differences in speed between S2 and CTR decrease and eventually change sign (Figure 2i). WiL is the only East Antarctic region where freshwater input is increased compared to CTR, but in spite of this the coastal current is losing speed. Therefore, both thermodynamic and dynamic response favour increased sea ice presence in the region.

In the BeS sector, the coastal current is least pronounced and current speeds are the lowest of all regions in CTR (Figure 2c). The S2 current speeds are even weaker in this region (Figure 2i). Sea ice drift therefore is of low importance. BeS is the only region where the local thermodynamic response clearly dominates the change in sea ice presence seen in S2.

### 3.1.3 S3: Response to wide-spread runoff addition

S3 features a widespread increase in sea ice concentration and thickness compared to CTR (Figure 2j-k), which is caused by higher local sea ice production. By decreasing the surface salinity, the enhanced runoff increases the freezing temperature and inhibits heat transport from below. Since in S3 only a part of the fresh water is added at the Antarctic shoreline, the coastal runoff is decreased compared to CTR in most areas and the coastal current is decelerated (Figure 2l), with the maximum deceleration along the Princess Martha Coast (Figure 1b), in the eastern Weddell Sea. Coastal velocities are faster than in CTR only from the Amundsen Sea to the Ross Ice Shelf front. In the Amundsen Sea, the increased speed leads to a sea ice depletion, because the ice is younger and the export from the region is increased. In the western Ross Sea, the increased velocities (Figure 3) lead to thicker sea ice (Figure 2, middle column) due to enhanced accumulation and compaction of the sea ice against the coastline in the southwestern corner of the Ross Sea. Additionally, a sea ice convergence is created by the contrast between the runoff addition at the southern and at the western coastline of the Ross Sea (Figure 1d) causing the ice drift to slow.

In the central and eastern Weddell Sea, the freshwater addition causes the ice to thicken thermodynamically in S3. In the western Weddell Sea, sea ice thickness is increased (Figure 2k), contrary to the ice concentration (Figure 2j). The increased sea ice presence over the northern part of the Weddell Gyre inhibits the northward export east of the Antarctic Peninsula (Figure 2l) and leads to dynamic compaction there.

### 3.1.4 S4: Response to regional runoff addition

The response of sea ice properties to modified runoff in S4 (Figure 2m-o) shows a similar pattern to that of S1 (Figure 2d-f). The strongest increase of both ice concentrations and thickness occurs around the tip of the Antarctic Peninsula and in the western Ross Sea, since the strengthened coastal current leads to more dynamical compaction in those areas. Thinner sea ice is found to the southeast of the peninsula (Figure 2n), which can be attributed to the fact that the ice is younger. Additionally, a decrease of sea ice concentration occurs at the Filchner/Ronne Ice Shelf front (Figure 2m). In this simulation, the extra runoff is regionally distributed and confined to the fronts of the Filchner/Ronne Ice Shelf (Weddell Sea), the Amery Ice Shelf (Prydz Bay) and the Ross Ice Shelf (Ross Sea). In the Weddell Sea, the coastal ice velocities increase (Figure 2o) and deplete the area of ice. In Prydz Bay, we find a similar speed increase and a local decrease of ice concentration. In the Ross Sea, the coastline geometry has a blocking effect on sea ice advected by the coastal current.

In summary, the sea ice response in our sensitivity experiments S1-S4 supports the hypothesis that artificial freshwater addition causes the sea ice to expand over time. The additional Antarctic runoff generally leads to an increase of the sea ice. Furthermore, our model results show that, in areas of a strong contrast in freshwater addition, the increase in drift velocity prevails against local thermodynamic effects and regionally, an enhanced freshwater supply can lead to thinner sea ice and lower sea ice concentrations. This is the expected case for the southern Weddell Sea and the Amundsen Sea, characterized by high mass loss and located downstream of areas where less runoff addition is expected. In regions located downstream of

large additional freshwater flux, where ocean velocities are increased, the sea ice can thicken due to enhanced dynamic compression, when encountering obstacles like headlands. This effect is evident in the western Ross Sea, but it occurs also at other locations, e.g. the tip of the Antarctic Peninsula in the Weddell Sea.

## 3.2 Development in time and variability

In this section, we assess the time-dependency of the effects of the extra freshwater input in the different experiments as well as its effect on the seasonal cycle of sea ice extent, volume and production. The time series of the sea ice properties are presented, together with their mean seasonal cycles, over the 10-year integration period in Figure 3.

The differences in sea ice extent (Figure 3a-b) between the sensitivity runs (S1-S4) and CTR are very small compared to the extent's seasonal amplitude (equal to $1.7 \times 10^7$ km$^2$), and present a marked interannual and seasonal variability. However, in all experiments, the additional fresh water enhances the monthly-averaged value of sea ice extent over the simulated period compared to CTR (Figure 3b). The S1 sea ice extent diverges from CTR only to a small degree, and, although the increase of ice extent prevails over the 10-year integration period, there are many occasions when S1 features a smaller sea ice extent than CTR. The magnitude of difference in ice extent between S2 and CTR is comparable with those of S1, but with distinct events of larger (smaller) ice extent in winter 2009-2011 (2012-2013).

S3 and S4 show a more substantial increase in sea ice extent. In S3, the widespread distribution of additional fresh water causes the sea ice to thermodynamically thicken and increases its concentration. In S4, the dominant factor is the dynamic compression due to more convergent ice drift. In both cases, the increased ice thickness lengthens the ice's lifespan. Therefore, the sea ice extent is increased and most effectively during the austral summer.

As for the sea ice extent, the differences of the sea ice volume between sensitivity runs and CTR (Figure 3c-d) are small compared to the volume's seasonal amplitude ($1.4 \times 10^4$ km$^3$). The S1 ice volume is generally comparable to CTR from February to May, but tends to increased values from June to January. The S2 differences to CTR in ice volume feature a larger interannual variability. During the first seven years, the volume generally surpasses that of CTR, but drops to lower values during 2011 to represent an increasing trend in the last two simulated years. In the 10-year mean, the seasonal cycle of S2 shows a larger volume than CTR, except in February and March. S3 produces higher sea ice volumes compared to CTR and all other experiments through almost the entire simulated period, due to the widespread increase in both sea ice concentration and thickness. Similar to S2, the initial strong increase is interrupted in 2011 when sea ice volume suddenly drops down, although remaining higher than CTR. These two experiments featuring a drop in ice volume in 2011 share a strongly regional distribution of runoff, suggesting that other regional factor may be in play, probably of atmospheric origin. The main contributing regions are the Amundsen, Bellinghausen and western Weddell Seas. S4, like S1, seems unaffected by the 2011 event and features distinctly increased ice volumes compared to CTR. The difference is comparatively small in the end of summer and reaches maximum values in spring.

Figures 3e-f show the changes in sea ice production caused by the runoff alterations. Again, the differences in sea ice production (Figure 3e) are small compared to the seasonal amplitude ($1.6 \times 10^6$ m$^3$ s$^{-1}$) of the ice production. All shown

experiments feature a sea ice production larger than CTR from autumn to spring, but in contrast the summer melting is also higher in S1-S4 than in CTR. In S1, the changes are the smallest and in magnitude comparable to their variability, while S3 diverges from CTR to the greatest extent and maintains a distinctly higher ice production even late in the year. While a strong stratification and a decoupled surface layer lead to cold surface waters and high ice production during the freezing period, in summer the heat uptake by the ocean is distributed in a shallower layer. Thus SST is higher and sea ice melt is enhanced. This behaviour is strengthened by a positive feedback loop (Stammerjohn et al., 2012) as long as the ocean gains heat.

## 4. Comparison with previous studies

Our numerical study primarily aims to investigate the response of sea ice to artificially-increased freshwater input and whether the ice response depends on the freshwater spatial distribution. Our sensitivity experiments S1-S4 have a higher amount of runoff compared to CTR that results in more sea ice. On a hemispheric scale, they confirm the expectation that an increase in Antarctic runoff leads to an increase in sea ice, in accordance with e.g. Bintanja et al. (2013), Bintanja et al. (2015) and Pauling et al. (2016).

The differences in fresh water applied in our simulations compared to CTR do not directly relate to the changes in Antarctic melt water estimated for the recent decades. An abrupt shift of freshwater sources from one region to another (as a comparison of CTR with S2 or S3 symbolizes) is unlikely. The increasing ocean temperatures are more likely to induce a slow (but region-dependent) increase of freshwater input.

Like Swart and Fyfe (2013), we find the simulated trends in sea ice extent to be smaller than the observed trend for runoff amounts close to observations and do not see the runoff as the main driving force of the circumpolar trend as found in simulations by Bintanja et al. (2013). However, we argue that the melt water increase currently contributes a roughly estimated 5 – 24 % of the observed increase in sea ice extent and is thus not negligible.

We find the spatial distribution of the freshwater addition of high influence on the sea ice cover, as Zunz and Goosse (2015) suspected. In particular, our results confirm the results of Merino et al. (2016) who showed that an idealized freshwater discharge from icebergs strongly impacts sea ice thickness, which in turn affects ice dynamics and longevity. Considering the effect that the spatial distribution of runoff has on sea ice, it seems important for modelling purposes to use a meltwater distribution as close to observations as possible. A re-adjustment of the sea ice parameters may be necessary to overcome the bias from tuning with a spatially unrealistic addition of fresh water.

Our three additional experiments, where the freshwater input (up to 550 Gt yr$^{-1}$) is beyond the range of current estimates, are an attempt to determine how an "extreme" amount of fresh water in our model configuration impacts the sea ice extent. The experiments S2-High and S3-High were motivated by the results of S2-Incr, with the objective of verifying and understanding the unexpected circumstance that more fresh water can cause a decrease in sea ice (as in the comparison of S2-Incr with S2). As described in Sect. 2.2, S2-Incr and S2-High differ from S2 in the amount of additional fresh water, but are based on the same spatial distribution, and S3-High features the same freshwater enhancement as S2-High, but with the

S3 spatial distribution that also mimics iceberg melt. Compared to S2 and S3, the amount of Antarctic fresh water is increased by 20 % in S2-High and S3-High (the 20 % increase is reached in 2012 in S2-Incr). Our model results suggest that the sea ice trend is dependent on the amount of fresh water added to the Southern Ocean. In fact, in response to "extreme" freshwater addition in our model, sea ice starts to decrease (Figure 4).

In all three experiments, there is a reduction of the sea ice extent and a loss of ice volume apparent in comparison with their respective base experiment (S2 or S3) toward the end of the simulated decade. The seasonal mean (Figure 4) still bears the imprint of a negative trend in both sea ice extent and volume in S2-High and S3-High. The ice loss occurs primarily in the Weddell Sea and is linked to a destabilisation of the water column. The faster coastal current, as dynamic response of the ocean to freshwater input (as in Figure 2), leads to increased Ekman pumping and offshore upwelling. Increased salinities

and temperatures at the surface can enhance ice melt or reduce ice formation. Once a reduction of sea ice occurs, a positive feedback loop between sea ice cover and ocean heat uptake from shortwave radiation is triggered (Stammerjohn et al., 2012). Our results with "extreme" freshwater input suggest that sea ice trend is sensitive to the amount of fresh water and to the method by which it is added. Pauling et al. (2016) performed experiments with freshwater addition larger than estimates (up to 3000 Gt yr$^{-1}$ increase) with a fully coupled model. Using two different freshwater distributions (ice shelf melt in front of

ice shelves and at the depth of the front, and iceberg melt at the surface over a wide area), they found that the total sea ice area increases significantly under the large freshwater enhancement. In accordance with the similarities we see between our experiments S2-High and S3-High after 10 years of integration, their experiments show that the spatial distributions of the freshwater input have no significant influence on the sea ice response. However, S2-High and S3-High feature divergent behaviour on seasonal time scales. As Pauling et al. (2016) point out, differences in the model complexity (as forced vs. fully

coupled configurations) and in their physics could lead to diverging results. With the low stability of the Southern Ocean water column, small differences in the chosen parameterizations, e.g. in the vertical mixing, can have a large effect on the sea ice. This is especially known for the Weddell Sea, which is the main region of sea ice loss in our experiments.

Of course, in contrast to our idealized experiments, the fresh water from ice sheet and iceberg basal melt does not enter the ocean only at the surface in the real world, but at tens or hundreds meters depth. However, this approximation, still widely-

25 used, is applied in our study. The effect on the sea ice may be small as Pauling et al. (2016) recently found for the depth distribution of additional fresh water in the Southern Ocean. Also, this study neglects the heat fluxes associated with the melting of glacial ice, which however makes it more comparable to the other studies.

## 5. On-shelf water characteristics

Sea ice and ocean are two components of the Earth's climate system that are strongly linked. Accounting for their complex

interplay is an important part of a precise reproduction and prediction of the climate. Sea ice formation and melt, which are strongly influenced by the availability of oceanic heat at the surface, directly affect the properties of the near-surface water masses. Sea surface salinity (SSS) can either be increased due to brine exclusion (during the formation of sea ice) or

decreased due to sea ice melting. Sea ice formation and brine exclusion rates play a vital role in the southern circulation regime and largely control the formation of the dense waters.

In this section, we present the effect of the freshwater additions in our experiments S1-S4 on the water columns in the key regions for dense water formation: the continental shelves of the Weddell Sea, the Ross Sea and the Prydz Bay. The areas are limited for the Weddell Sea to west of 38° W and south of 71.7° S, for the Ross Sea to west of 170° W and south of 71.7° S and for Prydz Bay to between 70° E and 80° E and south of 66.4° S. The regions are further limited to areas shallower than 550m depth to avoid strong influences from the deep ocean at the shelf break while including the outflow of dense water across the sills. Our comparison between S1-S4 and CTR focuses on the winter period (from April to September), when the dense shelf water is formed (Foldvik and Gammelsröd, 1988; Fahrbach et al., 1995).

In the mean vertical profiles of temperature, salinity and density computed in all three regions (Figure 5a-c, g-i, m-o), we find a warming (cooling) of the waters at 300-500m depth corresponding to the freshening (salinification) of the upper water column. Strong spatial variations in freshwater addition may cause local deviations from this behaviour, due to advection. For example, in the Weddell Sea, despite the local strong addition of fresh water, S2 features slightly saltier waters than CTR in the 100-300m depth interval, due to the preconditioning of the waters by the decreased runoff along the coastline of East Antarctica.

The evolution in time of the winter means of water properties at the 550m-isobath is presented for the simulated decade in Figure 5(d-f, j-l, p-r). The range of temperatures, salinities and densities at 550m depth between the simulations widens throughout the decade and the diverging trends can be expected to continue in subsequent years. The most extreme discrepancies of temperature and salinity in 2013 occur in the Ross Sea, where S2 features temperatures 1.2 K higher and salinities 0.09 psu higher than CTR. The highest discrepancy in density in 2013, however, occurs in the Weddell Sea, where the water in S3 is 0.06 kg m$^{-3}$ denser than in CTR.

In the Weddell Sea and in Prydz Bay, the freshwater distribution mimicking an iceberg drift pattern yields much cooler and consequently also denser shelf waters. The surface salinity of S3 is increased compared to CTR in these locations and the water column is destabilized, because the coastal freshwater input is reduced in the region upstream (East Antarctica). Only in very few locations, the coastal freshwater input of S3 is larger than that of CTR (Figure 1d). The most substantial increase of runoff occurs in the Amundsen Sea area, which is upstream of the Ross Sea shelf. Therefore, in S3, we see a subsurface warming and increased stability of the water column compared to CTR in the Ross Sea.

In contrast, S1 and S4 result in fresher shelf water than CTR in the Ross Sea. The freshening, however, mostly occurs in the last years of the simulated decade and is largely due to the destabilisation of the water column. Compared to CTR, S1 and S4 feature colder temperatures at 500m depth and higher salinities at the surface. Only S1 and S4 feature the same response in the dense shelf water in all three locations: temperatures are higher, salinities lower and densities decreased. In the Weddell and Ross Sea, S4 features the largest drop in salinity and consequentially density of all experiments. In Prydz Bay, S2 features a higher loss of salinity and density because here the water column is very unstable and the surface addition of fresh water easily translates to a freshening of the entire water column.

Our results regarding the formation of dense shelf water are in accordance with the findings of Hellmer (2004). Both studies support the idea that addition of fresh water leads to reduced density of the shelf waters, stronger stability of the water column and increased sea ice thickness (S1, S4). However, if aspects of spatially varying addition and subtraction of melt water come into play (as in S2, S3), the processes become more complex and the preconditioning of the waters in upstream regions can cause results to differ locally.

## 6. Conclusions

We have investigated the hypothesis that increasing freshening of the Southern Ocean could explain, at least partially, the Antarctic sea ice expansion. A set of coupled ocean-sea ice simulations with varying freshwater forcing were performed and compared to the control run in order to assess the impact of enhanced surface freshening on sea ice properties and dense water formation in the Southern Ocean. We used the NEMOv3.4 ocean model coupled with the LIM2 sea ice model in a global configuration with horizontal resolution of 1/4°.

Our results confirm that the sea ice extent (and volume) increases for moderate increases of the runoff amount. As also indic-ated in previous studies (e.g., Swingedouw et al. 2008, Bintanja et al, 2013), the enhanced freshwater input increases the near-surface stratification, which, in turn, inhibits the vertical transport of heat from depth to the ocean surface, a situation that can foster the formation of sea ice. However, we also find that, in our forced configuration, a large amount of freshwater can affect the sea ice trend inversely. Our experiments with the strongest freshwater forcing result in a decrease in sea ice ex-tent and volume.

For moderate enhancements of fresh water, the spatial distribution of the added water proves to be of great influence, since it affects the dynamic response of ocean and sea ice. For strong regional alterations of runoff addition, the dynamic response in our simulations is stronger than the thermodynamic response in most cases. The region with additional runoff is depleted of sea ice since the coastal current is accelerated, and sea ice export from the region increases. The spatial distribution of freshwater addition is therefore of great influence on the sea ice response.

Our model results show that the addition of freshwater can induce a warming in the sub-surface waters where the halocline and the near-surface stratification are strengthened and the vertical heat exchange are reduced. On the Antarctic continental shelves, the water characteristics are therefore subject to significant changes. In our experiments, the immediate and possibly transient response of the dense shelf water characteristics is a warming and freshening for simple increases in the runoff. The shelf waters hence become less dense. However, in regions downstream of reduced freshwater input at the coast, the water column is less stable and, in consequence, waters generated on the shelf are denser (colder and more saline).

We find that changes in the regional distribution of runoff can also induce regional variations in sea ice, as e.g. occurs in the Amundsen Sea, where the strong basal melt processes add a high amount of fresh water to the ocean. The dynamic response is an acceleration of coastal current and sea ice, which effectively reduces the sea ice cover, and exports more sea ice to the eastern Ross Sea. Generally, we conclude that the spatial distribution of runoff around the Antarctic continent is of high

importance for the sea ice cover and the stratification of the Southern Ocean water masses. Numerical applications may highly benefit from realistic distributions of Antarctic runoff.

It is worth noting that the impact on local ocean and sea ice, simulated in our experiments, is due to freshwater input that enters the ocean only through the surface. The sea ice response and the consequent impact on the water mass characteristics reproduced by our model may be sensitive to the depth of freshwater injection, and possibly be improved by better representation of the calving and basal melting of the ice shelves. The freshening of underlying layers would decrease stability and impact the mixed layer depth. Also the influence of the heat fluxes associated with melting the glacial ice has not been considered in this study.

## Acknowledgments

We thank the three reviewers and the editor for their thorough reading and constructive comments, which helped improving the manuscript. This study was performed in the framework of the "Climatically driven changes of Antarctic sea ice and their role in the climate system" (CATARSI) project as part of the Italian National Program for Research in Antarctica (PNRA). The financial support of the Italian Ministry of Education, University and Research, and Ministry for Environment, Land and Sea (MIUR), also through the project GEMINA, is gratefully acknowledged.

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

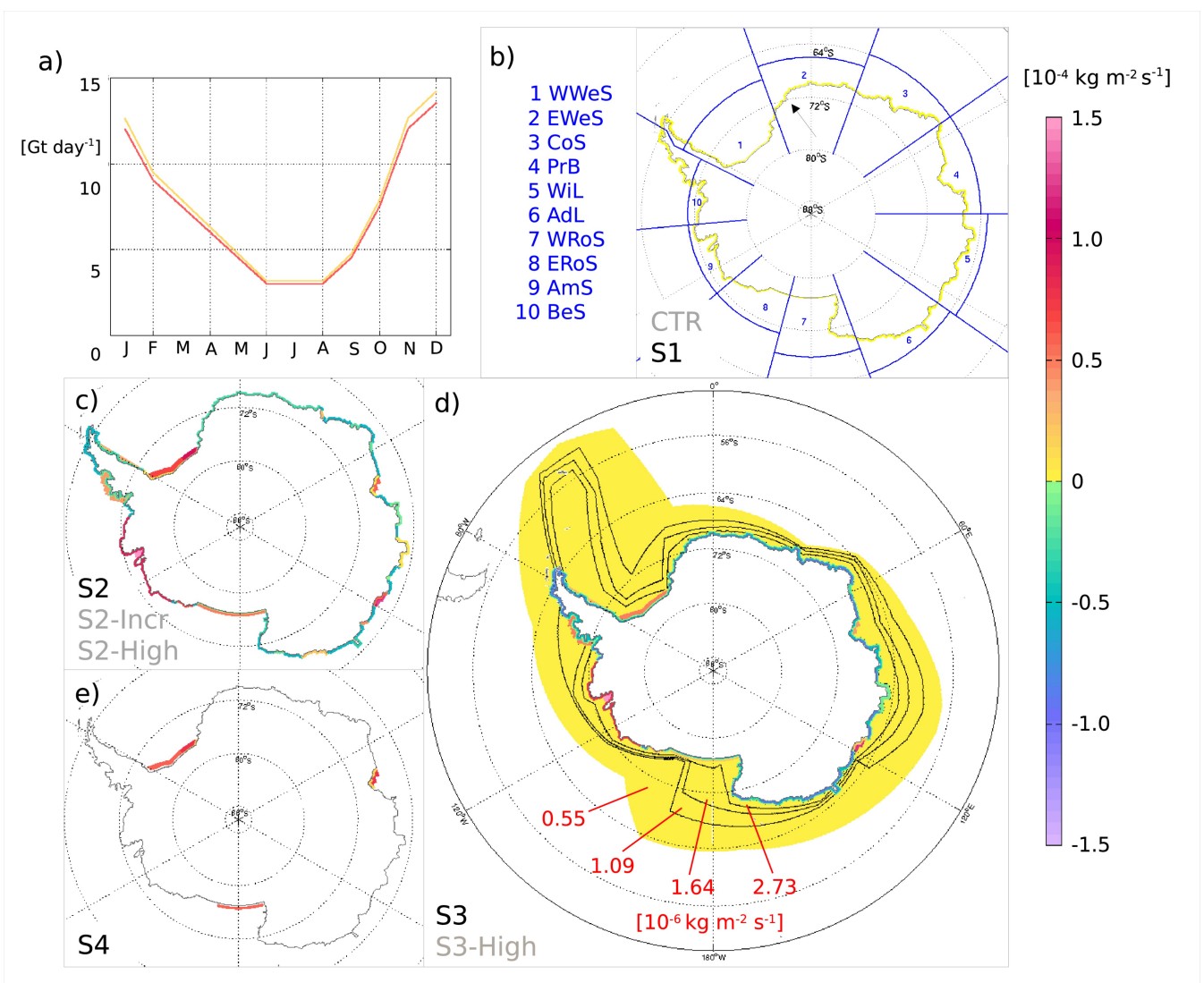

**Figure 1: a) Seasonal cycle of the runoff, exemplary for CTR (red) and S1 (yellow); b-e) Runoff distributions for the scenarios as differences from the CTR runoff. The numbers vary with the seasonal cycle; shown is the mean for the winter half of the year (April-September). CTR (S2-Incr, S2-High) has a similar distribution to S1 (S2) but different values; b) also depicts the location of Princess Martha coast marked with a black arrow and the definition of the regions used in the article: 1 WWeS – Western Weddell Sea, 2 EWeS – Eastern Weddell Sea, 3 CoS - Cosmonaut Sea, 4 PrB – Prydz Bay, 5 WiL – Wilkes Land, 6 AdL – Adelie Land, 7 WRoS – Western Ross Sea, 8 ERoS – Eastern Ross Sea, 9 AmS – Amundsen Sea, 10 BeS – Bellingshausen Sea.**

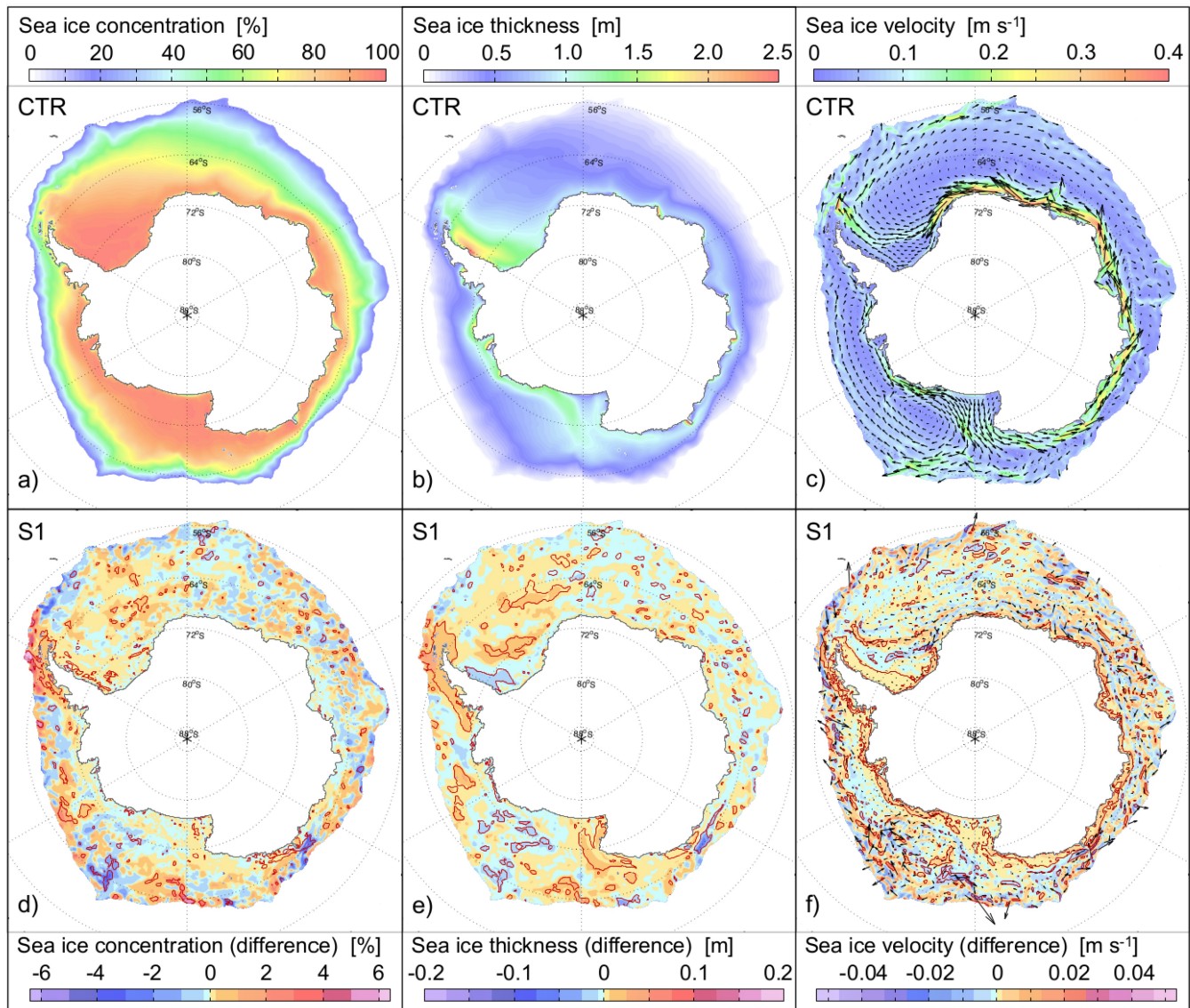

**Figure 2: Maps of winter sea ice a) concentration, b) thickness, and c) velocity in CTR averaged over April - September 2004 – 2013. d-o) Difference of ice concentration (left), thickness (middle), and velocity (right) between respective experiment and CTR. The colors underlying the velocity arrows indicate the difference in vector magnitude (speed). Dark red contours encompass the areas where the significance of the difference surpasses the 99 % confidence-level of the Student t-test for dependent samples, where the mean difference between two samples (here: time series) is set in relation with the standard deviation of the differences.**

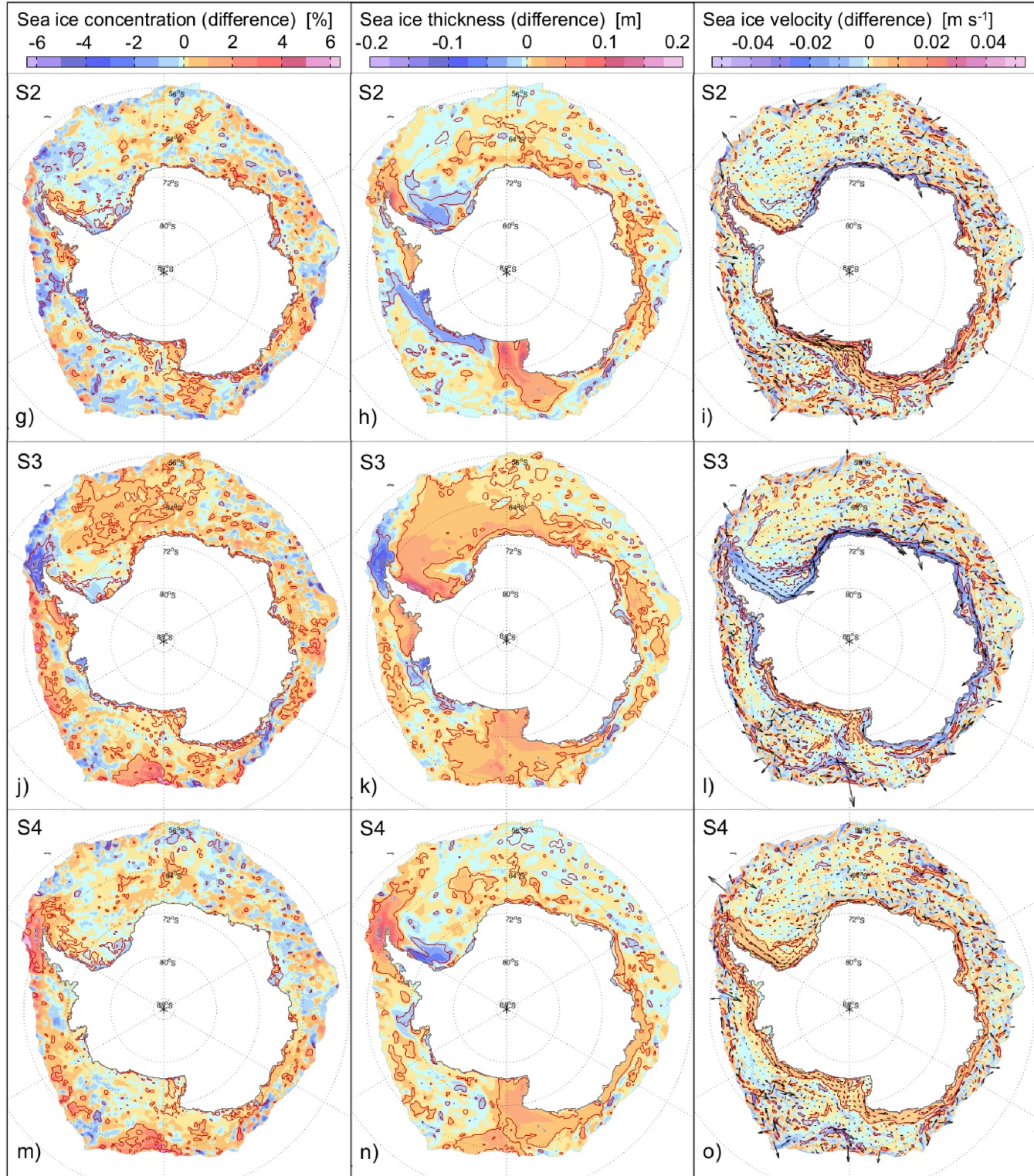

**Figure 2 (continued).**

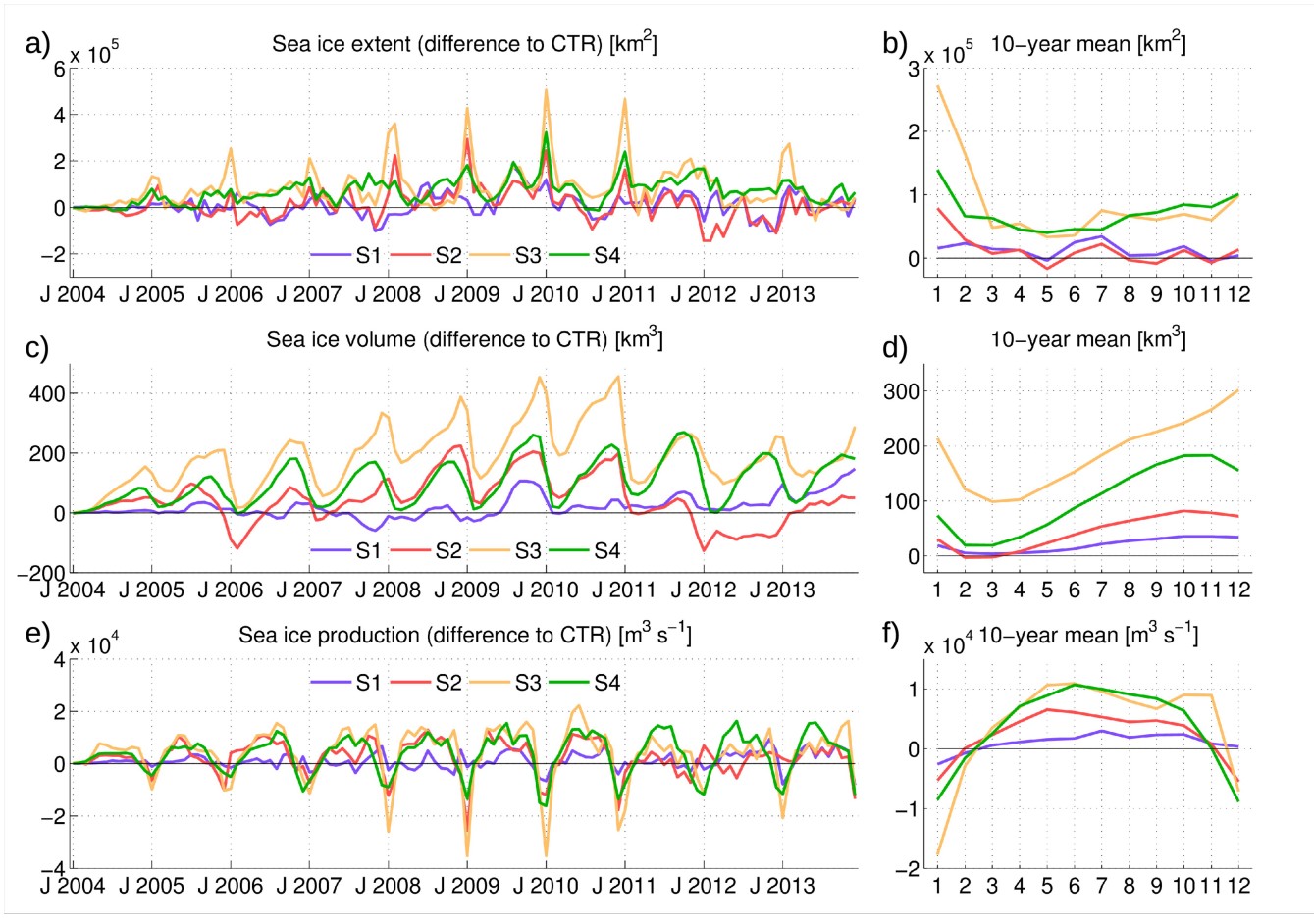

**Figure 3: Time series of the differences of (a-b) sea ice extent, (c-d) sea ice volume and (e-f) sea ice production between respective experiment and CTR, monthly values for 2004-2013 (left column), monthly values averaged over the 10-year period (right column).**

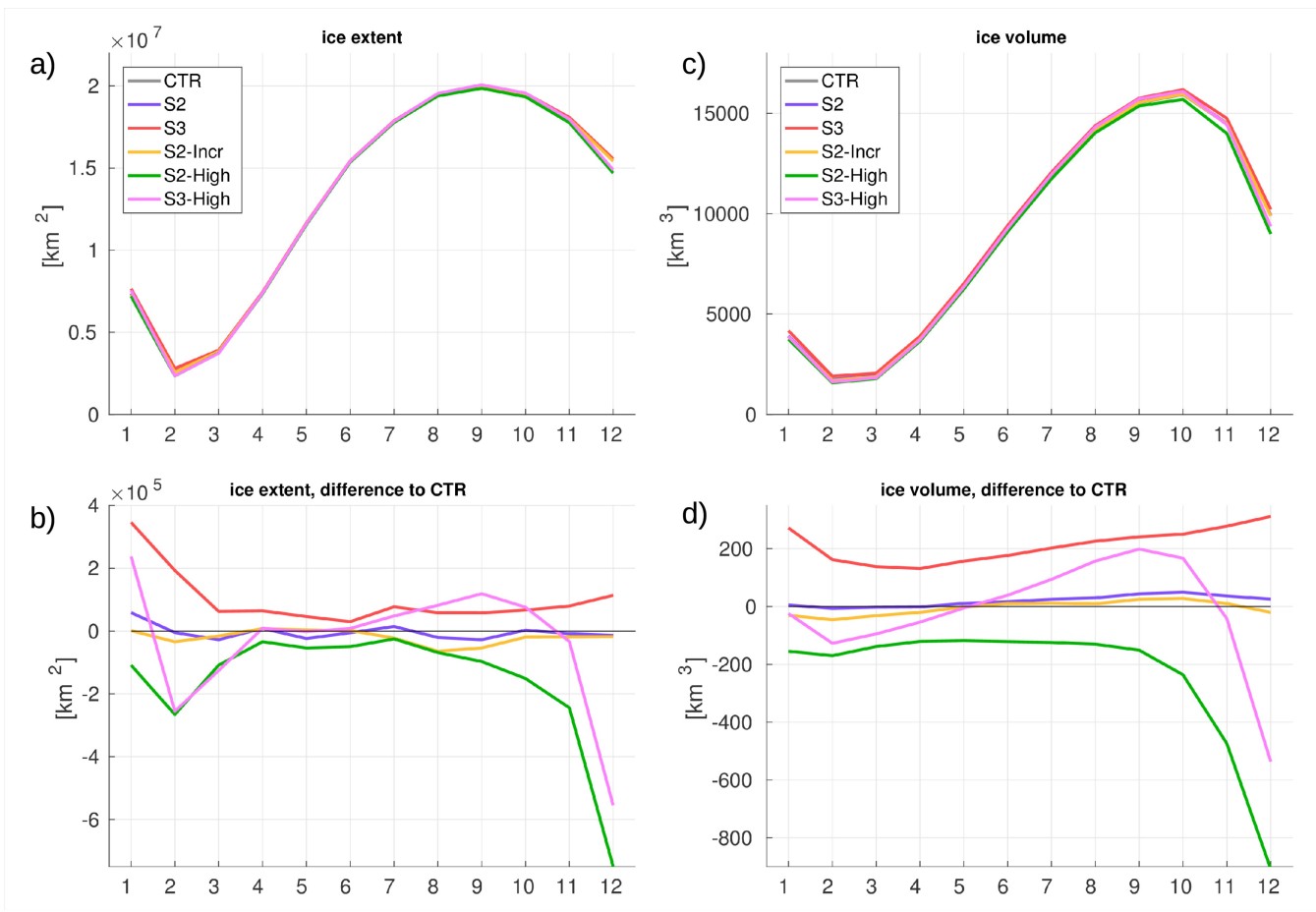

**Figure 4: Mean seasonal cycle of a) sea ice extent, b) differences to CTR in sea ice extent, c) sea ice volume and d) differences to CTR in sea ice volume for 2011-2013.**

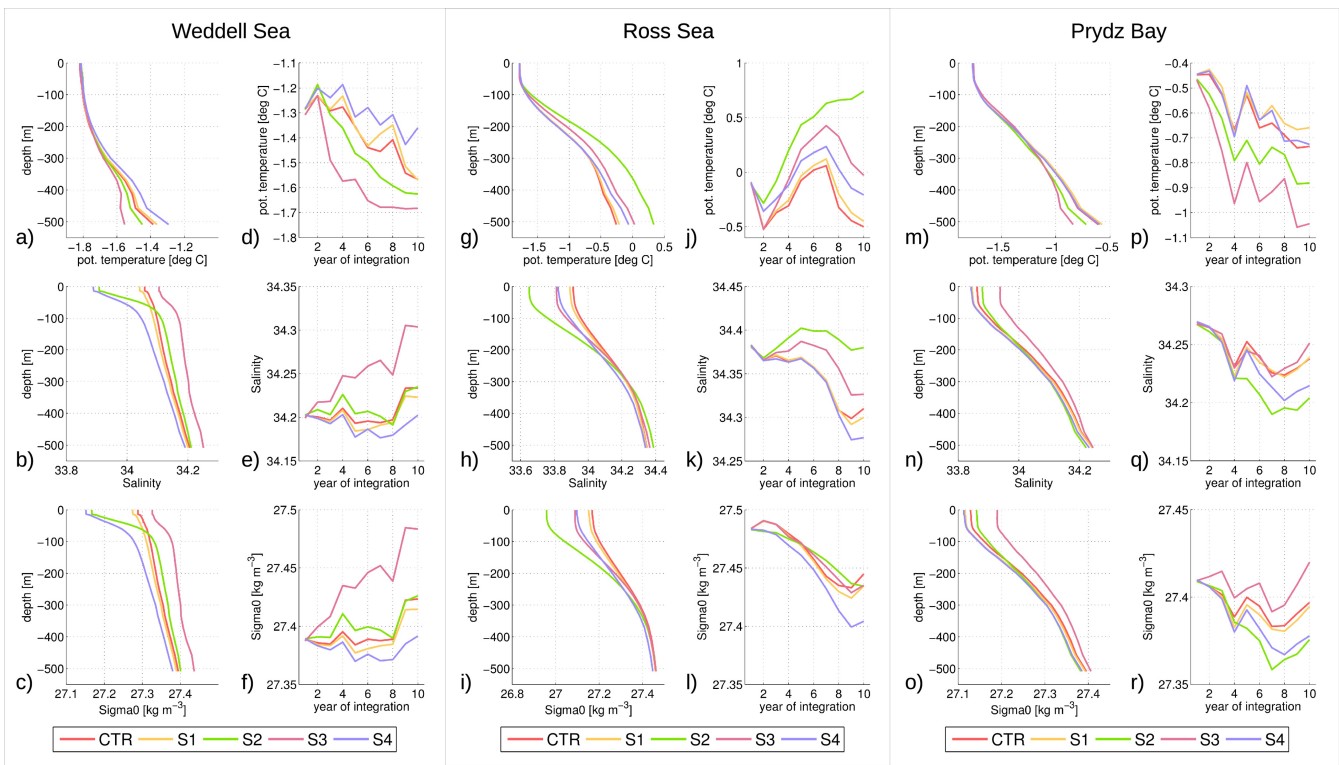

**Figure 5. Mean vertical profile ≦550 m depth (left) and annual mean values at the 550m isobath (right) of potential temperature, salinity and sigma0 in April-September in the a-f) Weddell Sea, g-l) Ross Sea and m-r) Prydz Bay.**

**Table 1: Main features of the experiments.**

| Simulation | Runoff amount [Gt yr$^{-1}$] | Runoff distribution |
|------------|------------------------------|---------------------|
| CTR | 2610 | Uniform, coastal |
| S1 | 2740 | Uniform, coastal |
| S2 | 2760 | Regional, coastal |
| S3 | 2760 | Regional, including offshore distribution |
| S4 | 3030 | Additional coastal runoff at major ice shelves |
| S2-Incr | 2760-3310 | Regional, coastal, increasing in 4 steps |
| S2-High | 3310 | Regional, coastal |
| S3-High | 3310 | Regional, including offshore distribution |

**Table 2. Differences between S2 and CTR computed as mean over the selected regions for April – September 2004 – 2013 period. Positive numbers are printed in bold font.**

| Region | Runoff [Mt] | Ice production [km$^3$ d$^{-1}$] | Ice concentration [%] | Ice thickness [cm] | Ice volume [km$^3$] |
|---|---|---|---|---|---|
| West. Weddell Sea (WWeS) | **30** | **0.17** | -0.1 | -1.0 | -14 |
| East. Weddell Sea (EWeS) | -8.4 | -0.07 | **0.04** | **0.6** | **5.0** |
| Cosmonaut Sea (CoS) | -16 | -0.05 | **0.3** | **1.1** | **6.8** |
| Prydz Bay (PrB) | -8.1 | -0.005 | **0.04** | **0.6** | **4.2** |
| Wilkes Land (WiL) | **1.1** | **0.02** | **0.02** | **0.8** | **4.0** |
| Adelie Land (AdL) | -13 | -0.06 | -0.06 | **0.3** | **2.0** |
| West. Ross Sea (WRoS) | -0.84 | **0.24** | **0.17** | **3.7** | **37** |
| East. Ross Sea (ERoS) | **8.4** | **0.15** | -0.16 | -1.6 | -10 |
| Amundsen Sea (AmS) | **42.7** | **0.15** | -0.34 | -3.3 | -11 |
| Bellingshausen Sea (BeS) | **0.9** | **0.016** | **0.29** | **1.8** | **2.8** |

