# Peer review of "Impacts of freshwater changes on the Antarctic sea ice in an eddypermitting sea ice-ocean model"

_The Cryosphere, 2016_

## Referee Comment (RC1) · R. Stevens (Referee) · 5 Jul 2016

R. Stevens (Referee)

roger.stevens@utas.edu.au

GENERAL COMMENTS Numerical models are the ideal tool for undertaking experiments on complex natural systems when attempting to understand the mechanics of the system and how it might change into the future. This paper presents the results of a set of numerical model experiments on one of Earth's major state changes, i.e. the freezing in autumn and winter of the Southern Ocean adjacent to the Antarctic Continent. What is more, Southern Ocean sea ice is baffling in that its extent has been increasing even as global air temperatures have increased. This paper investigates how much of the expansion of sea ice extent is caused by increased freshwater runoff from the Continent. The paper is therefore of interest not only for scientific reasons

but also because of the politics surrounding global warming and this "poster child" for climate change sceptics.

The paper is well written and easy to understand. The investigation into how the spatial distribution of fresh water input changes sea ice is interesting. That there could exist a maximum freshwater input for sea ice extent increase (and above which extent decreases) is also an interesting result.

SPECIFIC COMMENTS I think the experimental method is reasonable and the choice of NEMO/LIM is a good one. My major question relates to the authors' choice of LIM version 2 rather than version 3. I realize that LIM2 is favoured by ocean modellers because it is more economical on computer time but this research is focusing on sea ice (and the waters that interact with it) rather than the ocean generally. In this application LIM version 3 seems to offer some advantages over the earlier version. The most important of these is accounting for ice rafting and also frazil ice growth. Both of these processes are important for Southern Ocean sea ice (and less so for Arctic Ocean sea ice). The paper reports that one of the consequences of increased runoff is thinner, more mobile ice. Rafting is common, possibly ubiquitous, in thin Southern Ocean sea ice (Worby et al, 2008) and can occur in relatively mild convergent conditions compared to those encountered by sea ice impacting land, ice shelves, or land-fast ice. The results show that the most thickening of ice from ridging occurs in highly convergent regions, e.g. western Ross Sea. It is possible that the model underestimates dynamic ice thickening in other regions because of the lack of rafting in LIM2. It may have been interesting to run a simulation where increased freshwater is added to all the Southern Ocean, i.e. approximating increased precipitation. This would isolate the thermodynamic contribution of increased freshwater to the ice extent increase. However, I realize that these suggestions would require re-running the model and so are not feasible. It would be desirable to explain why LIM2 was preferred to LIM3. The simulated winter ice concentration is higher than that of satellite observations as seen in the supplementary material. The model will report high ice concentration of very

thin ice while the passive microwave observations will have problems distinguishing very thin ice from open water. However, thin ice also melts more quickly in spring and summer so I am not sure that the authors' argument is correct, i.e. that the higher winter ice concentration in the simulation accounts for the larger spring/early summer sea ice extent that the model produces. Extent includes open water south of the ice edge so maybe total ice area would give a better comparison? Using total ice area has its own problems of course. Also in the supplementary material the authors state that the quarter degree resolution is sufficient to capture most of the important aspects of atmosphere, ocean and thus the sea ice. I would agree with them in most respects but I wonder if they looked at how well the atmospheric forcing captured the katabatic winds which are so important for the formation of latent heat polynyas and therefore bottom water production?

TECHNICAL CORRECTIONS Poor grammar in places, e.g. page 8 lines 1: "In the central and eastern Weddell Sea, the fresh water addition causes the ice to thickened thermodynamically in S3." I think that "thickened" should be either "thicken" or "be thickened".

Worby, A.P., Geiger, Cathleen A., Paget, Matthew J., Van Woert, Michael L., Ackley, Stephen F., DeLiberty, Tracey L., 2008, "Thickness distribution of Antarctic sea ice", Journal of Geophysical Research, vol. 113 no. C05S92.

---

## Referee Comment (RC2) · Anonymous Referee #2 · 10 Jul 2016

This paper examines the model response of sea ice to the supply of additional freshwater at the surface of the ocean around Antarctica. The model used is NEMO, forced by global atmospheric reanalysis data with LIM2 sea ice model. Five scenarios are examined and compared with a control run. The scenarios include cases where the fresh water "runoff" is distributed uniformly around the coast of Antarctica, and others with regional maxima that approximately coincide with major ice shelves. In a third category the runoff is applied offshore, to mimic iceberg drift. The total magnitude of the runoff also differs between most of the simulations. The authors conclude that fresh water input increases sea ice extent and volume, up to a "turning point" value whereupon the sea ice trend is inverted. They also find that their experiments are sensitive to the

distribution of fresh water runoff at the ocean surface.

The paper is well written and readable and makes a useful contribution. One of the more interesting aspects of this paper is that the authors segregate the response of the sea ice into a thermodynamic and a dynamic components. I congratulate the authors on this part of their discussion.

MAIN COMMENTS

1. This is a topic of current interest, as evidenced by the fact that at least two highly relevant papers have appeared in the literature in the time that this article has been in process. Some details of the present paper need to acknowledge the publication of these two studies. They are

Merino, N. J. Le Sommer, G. Durand, N. Jourdain, G. Madec, P. Matthiot and J. Tournadre, (2016) Antarctic icebergs melt over the Southern Ocean: climatology and impact on sea-ice. Ocean Modelling, 104, 99-110, doi:10.1016/j.ocemod.2016.05.001

Pauling, A.G., C. M. Bitz, I. J. Smith, and P. J. Langhorne, (2016) The response of the Southern Ocean and Antarctic sea ice to fresh water from ice shelves in an Earth System Model. J. Climate, 29, 1655–1672. doi: http://dx.doi.org/10.1175/JCLI-D-15-0501.1

2. An interesting aspect is the hypothesis that a large amount of freshwater will reduce the sea ice. I am not sure I understand why this is the case. In addition, as the conclusion is based on one experiment, and as I could not see a clear pattern in the qualitative behaviour of the system with increasing freshwater flux, my opinion is that the authors need to work a little harder to be convincing.

3. In relation to this, please can you explain why the simulations of sea ice are considered to represent sea ice behaviour, while the simulation period of 10 years is too short for the water characteristics to reach equilibrium (see e.g. p. 2, line 28-33). Are you saying that you are investigating sea ice response processes and therefore do not

need to reach equilibrium? If this is the case, I am not sure I understand how you may conclude that there is a reversal of behaviour when more than a certain amount of fresh water (undetermined from these experiments) is added to the system. How can you tell that this is not due to variability between runs? This may require more explanation of the known behaviour of the model. The existence of a turning point based on evidence of a single simulation requires additional argument for its existence.

4. How was the seasonal variation in ice shelf "runoff" decided (see Fig 1e)?

5. Development in time and variability on p. 9: How much is know about variability between model runs when there has not been a repeat of an experiment? Perhaps this is well known for the model and could be briefly explained to the reader.

6. Comments 2-5 lead me to be unconvinced by the authors' conclusion that (the small) freshwater input they apply causes the sea ice to expand, while a larger input inverts the trend. This needs to be very carefully re-evaluated.

TECHNICAL COMMENTS

p. 2, line 9-10: Merino et al and Pauling et al (2016) need to be added to the previous studies.

p. 2, line 24-25: Note that Pauling et al (2016) have added fresh water spatially-distributed according to ice shelves, and at the depth of the ice shelf. However their simulations did not vary in magnitude through the year.

p. 2, line 28-33: (as main comment) Please can you explain why the simulations of sea ice are considered to represent sea ice behaviour, while the simulation period of 10 years is too short for the water characteristics to reach equilibrium. Are you saying that you are investigating sea ice response processes and therefore do not need to reach equilibrium? If this is the case, I am not sure I understand how you may conclude that there is a reversal of behaviour when more than a certain amount of fresh water (undetermined from these experiments) is added to the system. How can you tell that

need to reach equilibrium? If this is the case, I am not sure I understand how you may conclude that there is a reversal of behaviour when more than a certain amount of fresh water (undetermined from these experiments) is added to the system. How can you tell that this is not due to variability between runs? This may require more explanation of the known behaviour of the model. The existence of a turning point based on evidence of a single simulation requires additional argument for its existence.

4. How was the seasonal variation in ice shelf "runoff" decided (see Fig 1e)?

5. Development in time and variability on p. 9: How much is know about variability between model runs when there has not been a repeat of an experiment? Perhaps this is well known for the model and could be briefly explained to the reader.

6. Comments 2-5 lead me to be unconvinced by the authors' conclusion that (the small) freshwater input they apply causes the sea ice to expand, while a larger input inverts the trend. This needs to be very carefully re-evaluated.

TECHNICAL COMMENTS

p. 2, line 9-10: Merino et al and Pauling et al (2016) need to be added to the previous studies.

p. 2, line 24-25: Note that Pauling et al (2016) have added fresh water spatially-distributed according to ice shelves, and at the depth of the ice shelf. However their simulations did not vary in magnitude through the year.

p. 2, line 28-33: (as main comment) Please can you explain why the simulations of sea ice are considered to represent sea ice behaviour, while the simulation period of 10 years is too short for the water characteristics to reach equilibrium. Are you saying that you are investigating sea ice response processes and therefore do not need to reach equilibrium? If this is the case, I am not sure I understand how you may conclude that there is a reversal of behaviour when more than a certain amount of fresh water (undetermined from these experiments) is added to the system. How can you tell that

this is not due to variability between runs? This may require more explanation of the known behaviour of the model. The existence of a turning point based on evidence of a single simulation requires additional argument for its existence.

p. 4, line 5-6: Was Dai and Trenberth (2002) applied in all other parts of the globe, apart from Antarctica? Was the seasonal variation used (see Fig 1a – actually I think it is 1e) from Dai and Trenberth (2002)? If so how do you justify using the seasonal behaviour for river runoff to represent melting ice shelves?

p. 4: Table 1 is very useful but has not been referred to in the text. It would be useful to refer to it in section 2.2.

p. 4, lines 12-33: I think that the subfigures of Fig. 1 have been mislabeled.

p. 4-5: Experiment design – please note that Merino et al (2016) and Pauling et al (2016) both conduct experiments with fresh water distributed to mimic iceberg melt.

p. 5, line 20 onwards: This is a very interesting discussion regarding the influence of additional fresh water at the surface on the SSH, the velocity and thus on sea ice thickness. I was confused about how changes in the direction of the velocity were taken into account? Does the right hand column of Fig 2 show speed not velocity?

p. 5: Spatial Response Patterns: How can you have a high confidence interval in the difference when, at each time step, there are only two quantities? Is it time-averaged?

Fig 2 is for the "winter" months. Which months are "winter"?

p. 6: line 9-10: Is a salinity-dependent freezing point coded in the model?

p. 6, line 23 + p.7, lines 14, 27, 28, + p. 8, line 12, + p. 13 line 23: use of the word "acceleration" when I think you mean "faster speed".

p. 7, line 26: please mark Princess Martha Coast on a map.

p. 8, line 10: please mark Filchner/Ronne Ice Shelf on a map.

[Figure]

p. 9, line 10-12: Why are the larger amplitude anomalies in 2009-2011? Why are the anomalies smaller in 2012-2013?

p. 10, line 9-15: We are not shown the surface salinity or the SST so it is difficult to follow this discussion. Could the essential elements be presented in a figure?

p. 11, line 13 & line 16: I believe it is more appropriate to state as estimates 6 – 24% and 5 – 23% . p. 13, line 23: Stammerjohn et al (2012) have shown that there is strong correspondence between anomalies in the timings of sea ice retreat and subsequent advance, and you may wish to refer to this paper. Stammerjohn, S., R. Massom, D. Rind, and D. Martinson (2012), Regions of rapid sea ice change: An inter-hemispheric seasonal comparison, Geophys. Res. Lett., 39, L06501, doi:10.1029/2012GL050874. p. 13, line 23: Why would there be sea ice melt in winter? Is there evidence for this in the model runs? p. 14, line 2: replace "lose density" with "density reduces" p. 14, line 12-13: Some experiments have been done by Pauling et al (2016).

Fig 1: I did not understand the caption at all. I also think that the sub-figures are mislabeled. Please give a key for regions 1-10 in a).

Fig 2: Do you mean speed rather than velocity? What months are represented? How is the t-test performed when it is the difference between only 2 quantities?

Fig 3 b, d, f.: Are the large jumps in values between month 1 and month 12 expected?

---

## Author Response (AR1)

We would like to thank our two referees for their helpful comments and suggestions. We appreciate the time and effort that went into their review of our article and hope our explanations and actions taken upon their comments will satisfy the expectations of referees and editor.

Answers to Roger Stevens' comments:

GENERAL COMMENTS
Numerical models are the ideal tool for undertaking experiments on complex natural systems when attempting to understand the mechanics of the system and how it might change into the future. This paper presents the results of a set of numerical model experiments on one of Earth's major state changes, i.e. the freezing in autumn and winter of the Southern Ocean adjacent to the Antarctic Continent. What is more, Southern Ocean sea ice is baffling in that its extent has been increasing even as global air temperatures have increased. This paper investigates how much of the expansion of sea ice extent is caused by increased freshwater runoff from the Continent. The paper is therefore of interest not only for scientific reasons but also because of the politics surrounding global warming and this "poster child" for climate change sceptics. The paper is well written and easy to understand. The investigation into how the spatial distribution of fresh water input changes sea ice is interesting. That there could exist a maximum freshwater input for sea ice extent increase (and above which extent decreases) is also an interesting result.

SPECIFIC COMMENTS
I think the experimental method is reasonable and the choice of NEMO/LIM is a good one. My major question relates to the authors' choice of LIM version 2 rather than version 3. I realize that LIM2 is favoured by ocean modellers because it is more economical on computer time but this research is focusing on sea ice (and the waters that interact with it) rather than the ocean generally. In this application LIM version 3 seems to offer some advantages over the earlier version. The most important of these is accounting for ice rafting and also frazil ice growth. Both of these processes are important for Southern Ocean sea ice (and less so for Arctic Ocean sea ice). The paper reports that one of the consequences of increased runoff is thinner, more mobile ice. Rafting is common, possibly ubiquitous, in thin Southern Ocean sea ice (Worby et al, 2008) and can occur in relatively mild convergent conditions compared to those encountered by sea ice impacting land, ice shelves, or land-fast ice. The results show that the most thickening of ice from ridging occurs in highly convergent regions, e.g. western Ross Sea. It is possible that the model underestimates dynamic ice thickening in other regions because of the lack of rafting in LIM2. It may have been interesting to run a simulation where increased freshwater is added to all the Southern Ocean, i.e. approximating increased precipitation. This would isolate the thermodynamic contribution of increased freshwater to the ice extent increase. However, I realize that these suggestions would require re-running the model and so are not feasible. It would be desirable to explain why LIM2 was preferred to LIM3. The simulated winter ice concentration is higher than that of satellite observations as seen in the supplementary material. The model will report high ice concentration of very thin ice while the passive microwave observations will have problems distinguishing very thin ice from open water. However, thin ice also melts more quickly in spring and summer so I am not sure that the authors' argument is correct, i.e. that the higher winter ice concentration in the simulation accounts for the larger spring/early summer sea ice extent that the model produces. Extent includes open water south of the ice edge so maybe total ice area would give a better comparison? Using total ice area has its own problems of course. Also in the supplementary material the authors state that the quarter degree resolution is sufficient to capture most of the important

aspects of atmosphere, ocean and thus the sea ice. I would agree with them in most respects but I wonder if they looked at how well the atmospheric forcing captured the katabatic winds which are so important for the formation of latent heat polynyas and therefore bottom water production?

We agree with the referee that a more sophisticated sea ice model might have been more suitable for a model study focused on sea ice properties and their variability. We do also agree that the new features of LIM3 sea ice model might make it a better choice than LIM2, to a great extent. More than based on computational cost, our decision to remain on the "old" but well-tested LIM2 was motivated by the state of the LIM3 code at the time we started the research project that includes the simulations presented here (second half of 2014). Our modelling group at CMCC, as part of the NEMO system team, was aware of weaknesses of the LIM3 code available since 2009 (Vancoppenolle et al. 2009a,b) that might have a large impact on the Southern Ocean sea ice and water masses (as salt rejection during ice formation). The group closely followed the evolution of the new LIM code toward the updated version by Rousset et al. 2015. This new code has been released in July 2015, integrated into the most-recent stable version of NEMO (version 3.6). Tests, tuning and then analysis of the LIM3 performances in comparison to LIM2 have been conducted, starting from the coarse 1-degree global configuration (see Uotila et al., in review) and, only later, for the eddy-permitting configuration.
Following the development of the NEMO code, we do plan to continue our modeling study of the Antarctic sea ice and runoff effect using the more complete NEMO3.6 which includes the new LIM3 sea ice model, but also modules for iceberg and ice shelf cavities.

Rousset C., M. Vancoppenolle, G. Madec, T. Fichefet, S. Flavoni, A. Barthélemy, R. Benshila, J. Chanut, C. Levy, S. Masson, and F. Vivier: The Louvain-La-Neuve sea ice model LIM3.6: global and regional capabilities. Geosci. Model Dev., 8, 2991–3005, 2015. doi:10.5194/gmd-8-2991-2015

Uotila, P., Iovino, D., Vancoppenolle, M., Lensu, M., and Rousset, C.: On the influence of sea-ice physics in multi-decadal ocean-ice hindcasts, Geosci. Model Dev. Discuss., doi:10.5194/gmd-2016-187, in review, 2016.

Vancoppenolle, M., Fichefet, T., Goosse, H., Bouillon, S., Madec, G., and Maqueda, M. A.: Simulating the mass balance and salinity of Arctic and Antarctic sea ice: I. Model description and validation, Ocean Model., 27, 33–53, 2009a.

Vancoppenolle, M., Fichefet, T., and Goosse, H.: Simulating the mass balance and salinity of Arctic and Antarctic sea ice: II. Sensitivity to salinity processes, Ocean Model., 27, 54–69, 2009b

TECHNICAL CORRECTIONS Poor grammar in places, e.g. page 8 lines 1: "In the central and eastern Weddell Sea, the fresh water addition causes the ice to thickened thermodynamically in S3." I think that "thickened" should be either "thicken" or "be thickened".

We apologize for the mistakes. We carefully checked the text and hope to have corrected and eliminated all typographical and grammatical errors.

Answers to Anonymous Referee #2's comments :

This paper examines the model response of sea ice to the supply of additional freshwater at the surface of the ocean around Antarctica. The model used is NEMO, forced by global atmospheric reanalysis data with LIM2 sea ice model. Five scenarios are examined and compared with a control run. The scenarios include cases where the fresh water "runoff" is distributed uniformly around the coast of Antarctica, and others with regional maxima that approximately coincide with major ice shelves. In a third category the runoff is applied offshore, to mimic iceberg drift. The total magnitude of the runoff also differs between most of the simulations. The authors conclude that fresh water input increases sea ice extent and volume, up to a "turning point" value whereupon the sea ice trend is inverted. They also find that their experiments are sensitive to the distribution of fresh water runoff at the ocean surface. The paper is well written and readable and makes a useful contribution. One of the more interesting aspects of this paper is that the authors segregate the response of the sea ice into a thermodynamic and a dynamic components. I congratulate the authors on this part of their discussion.

MAIN COMMENTS
1. This is a topic of current interest, as evidenced by the fact that at least two highly relevant papers have appeared in the literature in the time that this article has been in process. Some details of the present paper need to acknowledge the publication of these two studies. They are
Merino, N. J. Le Sommer, G. Durand, N. Jourdain, G. Madec, P. Matthiot and J. Tournadre, (2016) Antarctic icebergs melt over the Southern Ocean: climatology and impact on sea-ice. Ocean Modelling, 104, 99-110, doi:10.1016/j.ocemod.2016.05.001
Pauling, A.G., C. M. Bitz, I. J. Smith, and P. J. Langhorne, (2016) The response of the Southern Ocean and Antarctic sea ice to fresh water from ice shelves in an Earth System Model. J. Climate, 29, 1655–1672. doi: http://dx.doi.org/10.1175/JCLI-D-15- 0501.1

We thank the referee for suggesting these new publications. We added the mentioned articles to the list of previous studies. Section 1. includes now the following text: "*Merino et al. (2016) used an iceberg model coupled to a sea ice-ocean model to establish a seasonal climatology of iceberg melt for the Southern Ocean and find that the iceberg melt water leads to higher sea ice concentration and thickness, with exception of the Amundsen/Bellingshausen Sea area. Pauling et al. (2016) employed an Earth-system model to investigate the Southern Ocean sea ice response to artificially augmented freshwater input. They tested the sensibility to higher freshwater additions than current estimates and previous studies and compare an iceberg model-based surface distribution with a coastal distribution at depth. They conclude that the effect of the different distributions on the mixed layer depth is contrary but the sea ice response is similar in both cases.*"

2. An interesting aspect is the hypothesis that a large amount of freshwater will reduce the sea ice. I am not sure I understand why this is the case. In addition, as the conclusion is based on one experiment, and as I could not see a clear pattern in the qualitative behaviour of the system with increasing freshwater flux, my opinion is that the authors need to work a little harder to be convincing.
We agree with the referee that the "turning point" conclusion based on only one experiment might be not solid enough. Therefore, we refrain from any statements presenting turning point as a fact. However, we find the results from that experiment very interesting and contrary to our expectations. This prompts us to show our findings and suggest a possible interpretation of them with the due caution.
As to the question why there may be a turning point, we offer a possible explanation at the end of

Section 3.2 p.10, which we have adapted after reconsidering the involved processes: *"During autumn the sea ice production of S5 surpasses that of S2, since the lower surface salinity facilitates ice formation. However, during winter and spring S2 features higher ice production values, because the influence of the offshore areas, particularly the northeastern Weddell and Ross Seas, becomes dominant. As a possible underlying mechanism, we suggest that the increased velocity is not limited to the coastal current but spreads to the subpolar gyres. A stronger circulation in a cyclonic gyre causes increased upwelling in the gyre's center due to the increased Ekman transport at the surface. In the Weddell and Ross Seas, this would cause a local increase of surface temperatures and salinities (SSS). In S5, SST and especially SSS in the winter mean is higher than in S2 in the northeastern Weddell Sea and northeastern Ross Sea (figures of the SST and SSS difference between experiments S2 and S5 are provided as Supplement S2). In consequence ice production is reduced and ice melt furthered. A reduced sea ice cover, especially in the regions close to the winter ice edge, leads to a higher heat uptake from solar radiation during the summer, triggering a positive feedback loop (Stammerjohn et al., 2012).*
*Additionaly, there is a second way, the increased speed of the coastal ice drift can contribute to the difference in sea ice volume and extent between S2 and S5: it shortens the period of time available for thermal growth and it can strengthen the mechanical processes thickening the ice in areas of convergence. Depending on the regional geometry and the ice drift pattern, either the thermodynamic or the dynamic effect on the sea ice thickness prevails and leads to thinner or thicker sea ice, respectively. While in WRoS, the sea ice in S5 is thicker than in S2 due to compression against the shoreline, the thermodynamic effect is of greater influence in WWeS, where large areas feature thinner ice in S5 (Figure 2h and q)."*
And summarize again in the Conclusions (Section 5):*"Based on this we think it probable that a turning point in the sea ice response to freshwater forcing exists and offer the following mechanism as a possible explanation: The coastal freshwater input changes the SSH slope and increases not only the velocities in the coastal current, but also of the subpolar gyres. Due to the increased Ekman transport more warm and saline water wells up in the gyres' centres., SST (and SSS) will increase and lead to enhanced melting of the northward advected sea ice and reduced local ice production during autumn and winter. The reduced sea ice cover allows higher shortwave radiation absorption by the ocean and triggers a positive feedback loop. Also, the freshwater-induced acceleration of the coastal current leads to thinner sea ice, when the time available for thermodynamical growth is reduced strongly. This is especially relevant for the Weddell Sea, while in the western Ross Sea all performed experiments result in dynamically thickened sea ice."*

3. In relation to this, please can you explain why the simulations of sea ice are considered to represent sea ice behaviour, while the simulation period of 10 years is too short for the water characteristics to reach equilibrium (see e.g. p. 2, line 28-33). Are you saying that you are investigating sea ice response processes and therefore do not need to reach equilibrium? If this is the case, I am not sure I understand how you may conclude that there is a reversal of behaviour when more than a certain amount of fresh water (undetermined from these experiments) is added to the system. How can you tell that this is not due to variability between runs? This may require more explanation of the known behaviour of the model. The existence of a turning point based on evidence of a single simulation requires additional argument for its existence.
The response times of ocean surface and deeper layers of the ocean to changes from the surface differ strongly. We are confident that the surface and the sea ice reaches near-equilibrium state within a fraction of the simulation period. We consider especially the differences between experiments to be reliable and due to the differences in runoff input since they present the only source of variability

between the runs. For these two reasons, we think the results trustworthy, although the ocean at depth is not in an equilibrium state.

With the complexity of today's ocean models and the short run period of our experiments it is possible that the difference we see between experiments S2 and S5 does not proof a turning point, however to the authors of this article the proposed explanation seems the most probable. The suggestion is not based on the result of one experiment, but on the difference we observe between two experiments. However, we agree with the referee that before stating the existence of this turning point as a fact further investigation is needed. We slightly altered the wording to enhance the speculative character of our suggestions.

4. How was the seasonal variation in ice shelf "runoff" decided (see Fig 1e)?

There is not much known about the seasonality of Antarctic runoff. However seasonality can be expected for both, iceberg meltwater and basal melting. In the first case the seasonality is strong (e.g. Merino et al., 2016) since the ocean surface heats up in summer. In the latter case uncertainties are large, but in winter the dense water formation in coastal polynyas inhibits warm water intrusions under the ice shelves and therefore a higher heat flux into the cavities can be expected during the summer. The runoff in our reference run is obtained from the DRAKKAR group (Bourdalle-Badie and Treguier 2006), who adapted the figures given by Dai and Trenberth (2002) and Jacobs et al. (1992) for the ORCA025 grid.

5. Development in time and variability on p. 9: How much is know about variability between model runs when there has not been a repeat of an experiment? Perhaps this is well known for the model and could be briefly explained to the reader.

The model variability is low and a repeat of any experiment is expected to give the same results, since the runtime of only ten years does not give the small numerical errors the time to grow into variability of any significance.

6. Comments 2-5 lead me to be unconvinced by the authors' conclusion that (the small) freshwater input they apply causes the sea ice to expand, while a larger input inverts the trend. This needs to be very carefully re-evaluated.

We agree that the presented experiments do not proof the existence of a turning point in the sea ice response. However, we are of the opinion that they strongly suggest such a behaviour. The differences in output between our experiment S2 and S5 are not random, but the result of the differences in the model input. We checked our wording to avoid misrepresentation of the turning point as a hard fact.

TECHNICAL COMMENTS

p. 2, line 9-10: Merino et al and Pauling et al (2016) need to be added to the previous studies.

We added Merino et al. (2016) and Pauling et al. (2016) to the previous studies. We added the following text in Section 1: "*Merino et al. (2016) used an iceberg model coupled to a sea ice-ocean model to establish a seasonal climatology of iceberg melt for the Southern Ocean and find that the iceberg melt water leads to higher sea ice concentration and thickness, with exception of the Amundsen/Bellingshausen Sea area. Pauling et al. (2016) employed an Earth-system model to investigate the Southern Ocean sea ice response to artificially augmented, constant freshwater input. They tested the sensibility to higher freshwater additions than current estimates and previous studies and compare an iceberg model-based surface distribution with a coastal distribution at depth. They conclude that the effect of the different distributions on the mixed layer depth is contrary but the sea ice*

*response is similar in both cases."*

We added the following passages to section 3.3: *" increase in Antarctic runoff leads to an increase in sea ice in accordance with e.g. Bintanja et al. (2013), Bintanja et al. (2015) and Pauling et al. (2016)."* *"While Pauling et al. (2016) with even higher amounts of fresh water addition did not conclude the existence of a turning point, their experiment with the highest amount of fresh water yields the lowest seasonal linear trends for the sea ice, while the lowest fresh water amount in summer and winter yields the least negative and in autumn even a positive trend (their Figure 11)."* *"In particular, as also Merino et al. (2016) found, considering an idealized freshwater discharge from icebergs strongly impacts sea ice thickness"* *"Pauling et al. (2016) recently found the depth distribution of additional fresh water in the Southern Ocean to be of small effect on the sea ice."*

p. 2, line 24-25: Note that Pauling et al (2016) have added fresh water spatially distributed according to ice shelves, and at the depth of the ice shelf. However their simulations did not vary in magnitude through the year.

Mentioned in the newly added text (see answer above).

p. 2, line 28-33: (as main comment) Please can you explain why the simulations of sea ice are considered to represent sea ice behaviour, while the simulation period of 10 years is too short for the water characteristics to reach equilibrium. Are you saying that you are investigating sea ice response processes and therefore do not need to reach equilibrium? If this is the case, I am not sure I understand how you may conclude that there is a reversal of behaviour when more than a certain amount of fresh water (undetermined from these experiments) is added to the system. How can you tell that this is not due to variability between runs? This may require more explanation of the known behaviour of the model. The existence of a turning point based on evidence of a single simulation requires additional argument for its existence.

Answered above (Main comments #2 and #3)

p. 4, line 5-6: Was Dai and Trenberth (2002) applied in all other parts of the globe, apart from Antarctica? Was the seasonal variation used (see Fig 1a – actually I think it is 1e) from Dai and Trenberth (2002)? If so how do you justify using the seasonal behaviour for river runoff to represent melting ice shelves?

The runoff data in the reference run is obtained from the DRAKKAR group (Bourdalle-Badie and Treguier, 2006) and is based on Dai and Trenberth (2002) for all the globe except Antarctica, where it relies on Jacobs et. al. (1992). The seasonal cycle of the Antarctic runoff was introduced by the DRAKKAR group. To more clearly explain this, we changed the description in the article to: *"The river run-off data is a monthly climatology based on the studies of Dai and Trenberth (2002) and Jacobs et al. (1992) and adapted for the ORCA025 grid by the DRAKKAR group (Bourdalle-Badie and Treguier, 2006)."* and added the references.

The melt of the Antarctic glacial ice in the Southern ocean is primarily dependent on the water temperature. For the basal melt of the ice shelves the main question is therefore how much warm water can intrude onto the continental shelves. In winter, deep convection linked to polynya activity hinders the warm water intrusions and thus higher melt rates can be expected in the summer months. Also for iceberg melt the seasonal dependence is strong due to the surface warming (Merino et al., 2016). We admit that there is limited knowledge of the seasonal cycle of the Antarctic 'runoff'. The runoff in our study therefore may not be correct in amplitude or shape, but some seasonal variation of the meltwater may be expected.

p. 4: Table 1 is very useful but has not been referred to in the text. It would be useful to refer to it in section 2.2.

*A reference was added in the text p.4, l. 16. "A short overview over the experiments and their differences is also given in Table 1."*

p. 4, lines 12-33: I think that the subfigures of Fig. 1 have been mislabeled.

*Yes, we apologize for the mix-up. The mistake is now corrected.*

p. 4-5: Experiment design – please note that Merino et al (2016) and Pauling et al (2016) both conduct experiments with fresh water distributed to mimic iceberg melt.

*In this section, we describe only our own experiments. However, both mentioned studies are now added with mention of the iceberg model-derived distribution in the introduction chapter.*

p. 5, line 20 onwards: This is a very interesting discussion regarding the influence of additional fresh water at the surface on the SSH, the velocity and thus on sea ice thickness. I was confused about how changes in the direction of the velocity were taken into account? Does the right hand column of Fig 2 show speed not velocity?

*The right hand column of Figure 2 shows the velocity differences as arrows underlaid by the difference in speed following the colour scale.*

p. 5: Spatial Response Patterns: How can you have a high confidence interval in the difference when, at each time step, there are only two quantities? Is it time-averaged?

*The confidence level was determined using the Student's t-test for dependent data samples. The 'sampling period' was limited to the 10\*6 monthly means April-September 2004-2013.*

Fig 2 is for the "winter" months. Which months are "winter"?

*We do refer to 'winter' as the 6-month period from April to September as mentioned p. 5, l. 14. A short statement was added there to improve clarity. "In the following, the word winter referring to a specific time period will mean the months April-September."*

p. 6: line 9-10: Is a salinity-dependent freezing point coded in the model?

*Yes, in the LIM2 sea ice model, the freezing temperature of seawater depends on salinity, linearly with an empirical constant.*

p. 6, line 23 + p.7, lines 14, 27, 28, + p. 8, line 12, + p. 13 line 23: use of the word "acceleration" when I think you mean "faster speed"

*Yes, not in all cases the words accelerated/acceleration were used in their proper sense. We corrected the phrasing where necessary.*

p. 7, line 26: please mark Princess Martha Coast on a map.

*Princess Martha Coast was marked on the map in Fig. 1b)*

p. 8, line 10: please mark Filchner/Ronne Ice Shelf on a map

*We consider the Filchner/Ronne Ice Shelf to be a well known feature of the Antarctic geography like e.g. the Ross Ice Shelf, Amundsen Sea and Antarctic Peninsula. We are afraid readers unfamiliar with the main features of the Antarctic geography will have to refer to a map from another source.*

p. 9, line 10-12: Why are the larger amplitude anomalies in 2009-2011? Why are the anomalies smaller in 2012-2013?

We cannot answer this question completely. As mentioned in the article, regional time series show that the difference occurs in the Amundsen, Bellingshausen and western Weddell Seas. We therefore assume that the atmospheric circulation features a regional shift in those years that has a much stronger effect on the experiments with regionally varied runoff than on the experiments with uniform coastal runoff.

p. 10, line 9-15: We are not shown the surface salinity or the SST so it is difficult to follow this discussion. Could the essential elements be presented in a figure?

Figures of the difference in SSS and SST between runs S2 and S5 are added as Supplements S2. In the article that fact is now mentioned: *"(figures of the SST and SSS difference between experiments S2 and S5 are provided as Supplement S2)"*.

p. 11, line 13 & line 16: I believe it is more appropriate to state as estimates $6 - 24\%$ and $5 - 23\%$ .

We agree and rounded the given percentages.

p. 13, line 23: Stammerjohn et al (2012) have shown that there is strong correspondence between anomalies in the timings of sea ice retreat and subsequent advance, and you may wish to refer to this paper. Stammerjohn, S., R. Massom, D. Rind, and D. Martinson (2012), Regions of rapid sea ice change: An inter-hemispheric seasonal comparison, Geophys. Res. Lett., 39, L06501, doi:10.1029/2012GL050874.

We thank the referee for the suggestion. The reference was added in the article: *"A reduced sea ice cover, especially in the regions close to the winter ice edge, leads to a higher heat uptake from solar radiation during the summer, triggering a positive feedback loop (Stammerjohn et al., 2012). "*

p. 13, line 23: Why would there be sea ice melt in winter? Is there evidence for this in the model runs?

If the SST is above the freezing point, sea ice melts. In the weakly stratified Southern Ocean heat can be transported to the surface with relative ease. In the Weddell Sea this has lead to the occurrence of the well-known Weddell polynya in the 1970s. Here, ice is advected northward into regions that still retain heat from the summer months. To improve understanding, the wording was changed. *"If the stratification of the offshore water column is increased, SST will increase during summer and lead to enhanced melting of the northward advected sea ice and reduced local ice production."*

p. 14, line 2: replace "lose density" with "density reduces"

The wording has been changed. *"the dense shelf waters become warmer, fresher and hence less dense."*

p. 14, line 12-13: Some experiments have been done by Pauling et al (2016).

The study by Pauling et al. 2016 has been added in the Introduction chapter and the section 3.3.

Fig 1: I did not understand the caption at all. I also think that the sub-figures are mislabeled. Please give a key for regions 1-10 in a).

Yes, the subplots were mislabelled. The mistake was corrected. A key for the regions was added in the figure caption.

Fig 2: Do you mean speed rather than velocity? What months are represented? How is the t-test performed when it is the difference between only 2 quantities?

In the right hand column the colour scale refers to speed, while the arrows depict velocity. A short explanatory text was added to the caption. *"The colors underlaying the velocity arrows indicate speed."* The 'winter' period in our article always refers to the months April-September. The t-test for dependent samples is performed on the time series of the two quantities. The 'sampling period' was limited to the 10*6 monthly means April-September 2004-2013.

Fig 3 b, d, f.: Are the large jumps in values between month 1 and month 12 expected?

They correspond to what is also visible in the time series in a), c), e) and are not beyond what is expected. The strong seasonal cycle of the runoff addition may play a role here and cause a stronger seasonal signal in the sea ice properties.

[revised manuscript text omitted]

Contents

Supplement S1 Control run

Here, we present results from the control run (CTR). To get an overview over the model's performance, we compare them to observations of sea ice (Cavalieri et al., 1996) and ocean temperature and salinity (Orsi and Whitworth, 2005). For the sea ice characteristics, the main focus is on the last decade (2004-2013), since it will be used for the comparison in Section 3. For the water characteristics, the model data from the year corresponding to the observation was used.

S1.1 Sea ice characteristics
The modeled sea ice concentration around Antarctica is compared with data obtained from passive microwave data sensors (Cavalieri et al., 1996) in Figure S1a-d. The model result matches the observations very well. It successfully reproduces the spatial distribution for both the minimum sea ice extent in March and the maximum sea ice extent in September. In March, a sea ice maximum is found in the western Weddell Sea. Also the Ross and Amundsen Seas feature large remnants of the ice cover, while less sea ice is found along the coastline of East Antarctica. However, the March ice concentration in the simulation tends to be lower than in the satellite data.

[Figure]

Figure S1:  a-d) Sea ice concentrations, mean 2004-2013; e) mean seasonal cycle of sea ice extent 1979-2013 and f) divergence from mean seasonal cycle of sea ice extent, SSM/I observations (blue) and CTR (red).

In September, the model features a sea ice outline that matches the observed shape very well. We find northward outcrops of the sea ice east of the South Sandwich Islands (approx. 27° W), west of the southward bend of the Southwest Indian Ridge (approx. 15° E), close to the Kerguelen Plateau (approx. 80° E), at the eastern margin of the d'Urville Sea (approx. 155° E), and east of the Pacific Antarctic Ridge (approx. 152° W). While the observed sea ice covered area is matched very well by the model, the sea ice concentrations in September are slightly overestimated.

The confines of the sea ice covered area in winter are defined by the Antarctic circumpolar current (ACC). The warm surface temperatures of the ACC melt the ice and thus inhibit sea ice expansion. The larger discrepancies between model and observation can therefore be mainly attributed to the resolution of the bathymetry, which steers the ocean currents.

A time series of the sea ice extent for both, simulation and observation is calculated from the sea ice concentration data using a threshold of 15%. The mean seasonal cycle of the sea ice extent (Figure 2e) shows that CTR very well captures amplitude and range of the seasonal changes. The extremes in summer and winter are very close to the observations and also during the freezing period, the simulated sea ice extent does not differ far from the observations. Only during the melting season, the simulation features a considerably higher sea ice extent than observed. This indicates a delay in the onset of sea ice melting that can be explained by the overestimation of ice concentrations in winter. It is compensated by much faster melting toward the end of summer, which is due to a shallower summer mixed layer of the model leading to an overestimation of ocean surface temperatures.

A time series of the deviations of the sea ice extent from the mean seasonal cycle (Figure 2f) features periods of large discrepancies during the first decade. Thereafter, the differences between CTR and observations in the sea ice extent are smaller and the short-term anomalies of the sea ice extent show similar fluctuations in CTR and in the satellite observations.

The initially high but later diminished discrepancies between simulation and observation indicate that after 10 years the model is close to equilibrium. In the period 1989-2013, the short-term fluctuations of the sea ice extent predominantly seem to be caused by similar events in model and observation. Although resolution remains an ever-present challenge in sea ice modeling, the 0.25° resolution of the model proves high enough to transmit the most important fluctuations of atmosphere and ocean to the sea ice.

The trend for the sea ice extent calculated for the period 1989-2013 from the simulation (31.000 $km^2$ $yr^{-1}$) matches the corresponding trend from the observations (25.000 $km^2$ $yr^{-1}$) fairly well. However, the simulation's trend decreases for later (shorter) periods and turns even slightly negative for the last decade. Since this behavior is not seen in the observations, the model is evidently missing an aspect of the development of the sea ice extent and the atmosphere (in the simulation) can not by itself account for the change. It is therefore a likely assumption that changes (increase) in the Antarctic runoff, a probable driving force behind the positive trend, are the missing piece in the puzzle.

For other sea ice characteristics there is still only sparse data available and a comprehensive comparison between model results and observations is not yet possible. However, the sea ice thickness of CTR (Figure 3a) is close to expectations. Typically the highest sea ice thicknesses are found close to the continent, while the areas farther offshore are covered with thinner ice. In the Weddell and Ross Seas, we can see the northward transport of the sea ice leaving traces in the thickness distribution. The coastal westward drift (Figure 3b) leads to dynamic build up of the sea ice thickness on the eastern flanks of the coastline while the western flanks are typically accompanied by a minimum in ice thickness.

The long-term mean sea ice velocities are strongly influenced by the ocean surface velocities. In the ocean, the westward coastal current following the continental shelf break is connected to the eastward ACC by the cyclonic subpolar gyres. In the Ross Sea, we find the Ross Gyre not only exporting sea ice northward on its western branch, but also causing southward drift of sea ice on its eastern branch, although to a much smaller extent. The Weddell Gyre has a less well-defined eastern, southward branch, due to its greater variability, and there is no southward ice drift visible in the long-term mean.

[Figure]

Figure S2: a) Mean sea ice thickness in CTR, April-September 2004-2013. b) Mean sea ice velocity in CTR, April-September 2004-2013.

S1.2 Water characteristics

In this chapter, salinity and potential temperature of CTR are compared to observations on two vertical sections through the Weddell and the Ross Seas. The measurements were taken during 1995 and 1992, respectively, and are compared to the annual mean of the corresponding year from the simulation.

The salinity and temperature section of CTR in the Weddell Sea from 15° W, 75° S to 37° W, 30° S is compared to the World Ocean Circulation Experiment (WOCE) Southern Ocean Atlas (Orsi and Whitworth, 2005) section A23. The comparison shows, that the model performs very well in reproducing the general characteristics of the different water masses. In the south close to the Antarctic continent, we find very cold and fresh waters at the surface with minimum temperatures close to the freezing point. Below the surface layer, we find slightly warmer and more saline waters, the Warm Deep Water (WDW), with a maximum in temperatures and salinity at ca. 400 m depth. With salinities slightly over 34.7 and temperatures reaching 1.4° C the simulation overestimates the maximum of the WDW. Below the WDW, salinity and temperature decrease again toward the bottom, where salinities below 34.66 and temperatures below -0.7° C are reached in the simulation. Compared to the observation the model's bottom values are slightly too high.

In the north, the surface waters are several degrees warmer than in the south and temperatures decrease with depth, reaching -0.4° C at the bottom. Salinities, however, increase with depth in the upper ocean and we find a maximum of over 34.76 at depths of 1500-2000 m, below which salinities slightly decrease again to less than 34.66 at the bottom.

The slight overestimation of salinity and temperature of the WDW in the Weddell Gyre can be attributed to the fact that the coarser model topography allows the waters of the ACC to mix more easily with those of the gyre over the South Scotia Ridge.

[Figure]

Figure S3: Section through the Weddell Sea (WOCE A23). a) CTR pot. temperature, b) difference to observed pot. temperature (Orsi and Whitworth, 2005), c) CTR salinity, d) difference to observed salinity (Orsi and Whitworth, 2005).

Also the bottom water in the Weddell Basin is slightly too warm and too saline. Apart from deficiencies in the topography of the South Scotia Ridge, that might let the dense water escape more easily into the open ocean, it is a common problem for OGCMs to correctly reproduce the modification of dense shelf water and ensuing formation of bottom water. While sinking down along the continental slope, the dense waters lose their characteristics too fast due to the model's limitations in resolution.

[Figure]

Figure S4: Section at 150° W (WOCE P16). a) CTR pot. temperature, b) difference to observed pot. temperature (Orsi and Whitworth, 2005), c) CTR salinity, d) difference to observed salinity (Orsi and Whitworth, 2005).

Another section of salinity and potential temperature in the eastern Ross Sea at 150° W also shows very good agreement between model and observation (WOCE section P16) (Figure 5). At the surface, we find cold and fresh (S ≈ 34.0) waters close to the Antarctic continent and warm, slightly saltier (S ≈ 34.5) waters in the north. At the bottom, we see the saline (S > 34.7) and relatively cold (T < 0.8° C) waters from Antarctica spill over the topographic obstacle of the Pacific Antarctic Ridge into the world ocean. In the simulation, the bottom waters retained in the southern basin close to the Antarctic continent with temperatures of approx. 0.9° C and salinities of approx. 34.7 are slightly warmer and more saline than in the observation. As in the Weddell Sea, two processes can contribute to this: First, the coarseness of the bathymetry in the model allows the dense water to escape more easily from the basin, and second, the resolution of the model hinders the production of the dense deep waters.

**Supplement S2 Difference between experiments S2 and S5 in SST and SSS**

[Figure]

Figure S5: Difference of 10-year mean winter sea surface temperatures between experiments S2 and S5.

[Figure]

Apr–Sep 2004–2013

Figure S6: Difference of 10-year mean winter sea surface salinities between experiments S2 and S5.

---

## Author Response (AR2)

We would like to thank our two referees for their helpful comments and suggestions. We appreciate the time and effort that went into their review of our article and hope our explanations and actions taken upon their comments will satisfy the expectations of referees and editor.

Answers to Roger Stevens' comments:

**GENERAL COMMENTS**

Numerical models are the ideal tool for undertaking experiments on complex natural systems when attempting to understand the mechanics of the system and how it might change into the future. This paper presents the results of a set of numerical model experiments on one of Earth's major state changes, i.e. the freezing in autumn and winter of the Southern Ocean adjacent to the Antarctic Continent. What is more, Southern Ocean sea ice is baffling in that its extent has been increasing even as global air temperatures have increased. This paper investigates how much of the expansion of sea ice extent is caused by increased freshwater runoff from the Continent. The paper is therefore of interest not only for scientific reasons but also because of the politics surrounding global warming and this "poster child" for climate change sceptics. The paper is well written and easy to understand. The investigation into how the spatial distribution of fresh water input changes sea ice is interesting. That there could exist a maximum freshwater input for sea ice extent increase (and above which extent decreases) is also an interesting result.

**SPECIFIC COMMENTS**

I think the experimental method is reasonable and the choice of NEMO/LIM is a good one. My major question relates to the authors' choice of LIM version 2 rather than version 3. I realize that LIM2 is favoured by ocean modellers because it is more economical on computer time but this research is focusing on sea ice (and the waters that interact with it) rather than the ocean generally. In this application LIM version 3 seems to offer some advantages over the earlier version. The most important of these is accounting for ice rafting and also frazil ice growth. Both of these processes are important for Southern Ocean sea ice (and less so for Arctic Ocean sea ice). The paper reports that one of the consequences of increased runoff is thinner, more mobile ice. Rafting is common, possibly ubiquitous, in thin Southern Ocean sea ice (Worby et al, 2008) and can occur in relatively mild convergent conditions compared to those encountered by sea ice impacting land, ice shelves, or land-fast ice. The results show that the most thickening of ice from ridging occurs in highly convergent regions, e.g. western Ross Sea. It is possible that the model underestimates dynamic ice thickening in other regions because of the lack of rafting in LIM2. It may have been interesting to run a simulation where increased freshwater is added to all the Southern Ocean, i.e. approximating increased precipitation. This would isolate the thermodynamic contribution of increased freshwater to the ice extent increase. However, I realize that these suggestions would require re-running the model and so are not feasible. It would be desirable to explain why LIM2 was preferred to LIM3. The simulated winter ice concentration is higher than that of satellite observations as seen in the supplementary material. The model will report high ice concentration of very thin ice while the passive microwave observations will have problems distinguishing very thin ice from open water. However, thin ice also melts more quickly in spring and summer so I am not sure that the authors' argument is correct, i.e. that the higher winter ice concentration in the simulation accounts for the larger spring/early summer sea ice extent that the model produces. Extent includes open water south of the ice edge so maybe total ice area would give a better comparison? Using total ice area has its own problems of course. Also in the supplementary material the authors state that the guarter degree resolution is sufficient to capture most of the important aspects of atmosphere, ocean and thus the sea ice. I would agree with them in most respects but I wonder if they looked at how well the atmospheric forcing captured the katabatic winds which are so important for the formation of latent heat polynyas and therefore bottom water production?

We agree with the referee that a more sophisticated sea ice model might have been more suitable for a model study focused on sea ice properties and their variability. We do also agree that the new features of LIM3 sea ice model might make it a better choice than LIM2, to a great extent. More than based on computational cost, our decision to remain on the "old" but well-tested LIM2 was motivated by the state of the LIM3 code at the time we started the research project that includes the simulations presented here (second half of 2014). Our modelling group at CMCC, as part of the NEMO system team, was aware of weaknesses of the LIM3 code available since 2009 (Vancoppenolle et al. 2009a,b) that might have a large impact on the Southern Ocean sea ice and water masses (as salt rejection during ice formation). The group closely followed the evolution of the new LIM code toward the updated version by Rousset et al. 2015. This new code has been released in July 2015, integrated into the most-recent stable version of NEMO (version 3.6). Tests, tuning and then analysis of the LIM3 performances in comparison to LIM2 have been conducted, starting from the coarse 1-degree global configuration (see Uotila et al., in review) and, only later, for the eddy-permitting configuration.

Following the development of the NEMO code, we do plan to continue our modeling study of the Antarctic sea ice and runoff effect using the more complete NEMO3.6 which includes the new LIM3 sea ice model, but also modules for iceberg and ice shelf cavities.

Rousset C., M. Vancoppenolle, G. Madec, T. Fichefet, S. Flavoni, A. Barthélemy, R. Benshila, J. Chanut, C. Levy, S. Masson, and F. Vivier: The Louvain-La-Neuve sea ice model LIM3.6: global and regional capabilities. Geosci. Model Dev., 8, 2991–3005, 2015. doi:10.5194/gmd-8-2991-2015

Uotila, P., Iovino, D., Vancoppenolle, M., Lensu, M., and Rousset, C.: On the influence of sea-ice physics in multi-decadal ocean-ice hindcasts, Geosci. Model Dev. Discuss., doi:10.5194/gmd-2016-187, in review, 2016.

Vancoppenolle, M., Fichefet, T., Goosse, H., Bouillon, S., Madec, G., and Maqueda, M. A.: Simulating the mass balance and salinity of Arctic and Antarctic sea ice: I. Model description and validation, Ocean Model., 27, 33–53, 2009a.

Vancoppenolle, M., Fichefet, T., and Goosse, H.: Simulating the mass balance and salinity of Arctic and Antarctic sea ice: II. Sensitivity to salinity processes, Ocean Model., 27, 54–69, 2009b

TECHNICAL CORRECTIONS Poor grammar in places, e.g. page 8 lines 1: "In the central and eastern Weddell Sea, the fresh water addition causes the ice to thickened thermodynamically in S3." I think that "thickened" should be either "thicken" or "be thickened".

We apologize for the mistakes. We carefully checked the text and hope to have corrected and eliminated all typographical and grammatical errors.

Answers to Anonymous Referee #2's comments :

This paper examines the model response of sea ice to the supply of additional freshwater at the surface of the ocean around Antarctica. The model used is NEMO, forced by global atmospheric reanalysis data with LIM2 sea ice model. Five scenarios are examined and compared with a control run. The scenarios include cases where the fresh water "runoff" is distributed uniformly around the coast of Antarctica, and others with regional maxima that approximately coincide with major ice shelves. In a third category the runoff is applied offshore, to mimic iceberg drift. The total magnitude of the runoff also differs between most of the simulations. The authors conclude that fresh water input increases sea ice extent and volume, up to a "turning point" value whereupon the sea ice trend is inverted. They also find that their experiments are sensitive to the distribution of fresh water runoff at the ocean surface. The paper is well written and readable and makes a useful contribution. One of the more interesting aspects of this paper is that the authors segregate the response of the sea ice into a thermodynamic and a dynamic components. I congratulate the authors on this part of their discussion.

**MAIN COMMENTS**

1. This is a topic of current interest, as evidenced by the fact that at least two highly relevant papers have appeared in the literature in the time that this article has been in process. Some details of the present paper need to acknowledge the publication of these two studies. They are

Merino, N. J. Le Sommer, G. Durand, N. Jourdain, G. Madec, P. Matthiot and J. Tournadre, (2016) Antarctic icebergs melt over the Southern Ocean: climatology and impact on sea-ice. Ocean Modelling, 104, 99-110, doi:10.1016/j.ocemod.2016.05.001

Pauling, A.G., C. M. Bitz, I. J. Smith, and P. J. Langhorne, (2016) The response of the Southern Ocean and Antarctic sea ice to fresh water from ice shelves in an Earth System Model. J. Climate, 29, 1655–1672. doi: http://dx.doi.org/10.1175/JCLI-D-15-0501.1

We thank the referee for suggesting these new publications. We added the mentioned articles to the list of previous studies. Section 1. includes now the following text (p. 2, l. 32 - p. 3, l. 5): "Merino et al. (2016) used an iceberg model coupled to a sea ice-ocean model to establish a seasonal climatology of iceberg melt for the Southern Ocean. They find that the iceberg melt water leads to higher sea ice concentration and thickness, with exception of the Amundsen/Bellingshausen Sea area. Pauling et al. (2016) employed an Earth-system model to investigate the effect of artificially augmented, constant freshwater input on sea ice. They tested the sensibility to freshwater additions from current estimates to much larger values (3000 Gt yr-1) and compared an iceberg model-based surface distribution with a coastal distribution at the depth of the ice shelf front. Their model experiments show that enhancing the freshwater input by an amount within the range of estimates of the Antarctic mass imbalance does not significantly affect sea ice area, magnitude or trend. Further, their results show the sea ice response to be insensitive to the depth of freshwater injection."

In Section 4, we compare our findings with their results:

(p. 11, l. 24) "On a hemispheric scale, they confirm the expectation that an increase in Antarctic runoff leads to an increase in sea ice, in accordance with e.g. Bintanja et al. (2013), Bintanja et al. (2015) and Pauling et al. (2016)."

(p. 12, l. 20) "In particular, as also Merino et al. (2016) found, considering an idealized freshwater discharge from icebergs strongly impacts sea ice thickness, which in turn affects ice dynamics and longevity."

(p. 13, l. 9-16) "Pauling et al. (2016) performed experiments with freshwater addition larger than estimates (up to 3000 Gt yr-1 increase) with a fully coupled model. Using two different freshwater

distributions (ice shelf melt in front of ice shelves and at the depth of the front, and iceberg melt at the surface over a wide area), they found that the total sea ice area increases significantly under the large freshwater enhancement. In accordance with the similarities we see between our experiments S2-High and S3-High after 10 years of integration, their experiments show that the spatial distributions of the freshwater input have no significant influence on the sea ice response. However, S2-High and S3-High feature divergent behaviour on seasonal time scales. As Pauling et al. (2016) point out, differences in the model complexity (as forced vs. fully coupled configurations) and in their physics could lead to diverging results."

(p. 13, l. 21) "The effect on the sea ice may be small as Pauling et al. (2016) recently found for the depth distribution of additional fresh water in the Southern Ocean."

2. An interesting aspect is the hypothesis that a large amount of freshwater will reduce the sea ice. I am not sure I understand why this is the case. In addition, as the conclusion is based on one experiment, and as I could not see a clear pattern in the qualitative behaviour of the system with increasing freshwater flux, my opinion is that the authors need to work a little harder to be convincing.

We agree with the referee that the "turning point" conclusion based on only one experiment might be not solid enough. Since we find the results from that experiment very interesting and contrary to our expectations, we conducted two more simulations with larger freshwater addition in two different spatial distributions. The sea ice response in these runs strengthens the robustness of our model results. Under these 2 "extreme" runoff forcings, our model reproduces the same principal result supporting the hypothesis that sea ice extent and volume can decrease despite an increased freshwater addition.

The revised manuscript includes the new experiments, with consequent changes to the text.

The name S2-Incr is now used to identified the run previously-called S5. The new experiments with larger freshwater addition are named S2-High and S3-High and are based on runoff spatial distribution of S2 and S3, respectively. Figure 1 now includes the new experiments. We extracted S5 (now S2-Incr) from the Figures 2, 3 and 5 (formerly Figure 4) and included a new Figure 4, setting the ice extent and volume of the new runs in context with CTR, S2 and S3. Also, we changed the text at several locations, the most important of which follow:

In the Abstract (p. 1, l. 15):

"Only for "extreme" freshwater additions, our experiments show a negative development of the sea ice extent."

In Section 1 (p. 3, l. 18-21):

"Three additional experiments with enhanced freshwater amounts exceeding the range of current estimates (70-290 Gt yr-1; (Rignot et al., 2008; Joughin and Alley, 2011; Shepherd et al., 2012; Vaughan et al., 2013; Wouters et al., 2013; Rignot et al., 2013; Velicogna et al., 2014), allow us to gain insight in possible future sea ice changes."

In Section 2.1 (p. 4, l. 20):

"A suite of simulations is presented in this manuscript: a control run (hereafter CTR) and seven sensitivity experiments (S1-S4, S2-Incr, S2-High and S3-High)."

In Section 2.2 (p. 6, l. 7-12):

"We present three additional experiments in which the amount of additional fresh water exceeds the range of current Antarctic mass loss estimates. These runs are an attempt to determine how much fresh water is required to have a significant effect on the sea ice area trend, in our model. The spatial runoff distribution in the experiments S2-Incr and S2-High is based on S2 simulation, while the experiment S3-High follows the S3 distribution including the widespread offshore freshwater addition. In S2-Incr, the runoff increases from 2760 Gt yr-1 in 2004 to 3310 Gt yr-1 in 2013 in 4 steps (137 Gt yr-1 every 2 years). In S2-High and S3-High, a constant freshwater input of 3310 Gt yr-1 is added around Antarctica

in different spatial distributions."

In Section 3.1 and 3.2, we removed the paragraphs referring to S5.

We turned Section 3.3 into an independent Section 4, where we added the following text (p. 12, l. 25 – p. 13, l. 18):

"Our three additional experiments, where the freshwater input (up to 550 Gt yr-1) is beyond the range of current estimates, are an attempt to determine how an "extreme" amount of fresh water in our model configuration impacts the sea ice extent. The experiments S2-High and S3-High were motivated by the results of S2-Incr, with the objective to verify and understand the unexpected circumstance that more fresh water can cause a decrease in sea ice (as in the comparison of S2-Incr with S2). As described in Sect. 2.2, S2-Incr and S2-High differ from S2 in the amount of additional fresh water, but are based on the same spatial distribution, and S3-High features the same freshwater enhancement as S2-High, but with the S3 spatial distribution that also mimics iceberg melt. Compared to S2 and S3, the amount of Antarctic fresh water is increased by 20% in S2-High and S3-High (the 20% increase is reached in 2012 in S2-Incr). Our model results suggest that the sea ice trend is dependent on the amount of fresh water added to the Southern Ocean. In fact, in response to "extreme" freshwater addition in our model, sea ice starts to decrease (Figure 5).

In all three experiments, there is a reduction of the sea ice extent and a loss of ice volume apparent in comparison with their respective base experiment (S2 or S3) toward the end of the simulated decade. The seasonal mean (Figure 5) still bears the imprint of a negative trend in both sea ice extent and volume in S2-High and S3-High. The ice loss occurs primarily in the Weddell Sea and is linked to a destabilisation of the water column. The faster coastal current, as dynamic response of the ocean to freshwater input (as in Figure 2), leads to increased Ekman pumping and offshore upwelling. Increased salinities and temperatures at the surface can enhance ice melt or reduce ice formation. Once a reduction of sea ice occurs, a positive feedback loop between sea ice cover and ocean heat uptake from shortwave radiation is triggered (Stammerjohn et al., 2012).

Our results with "extreme" freshwater input suggest that sea ice trend is sensitive to the amount of fresh water and to the method by which it is added. Pauling et al. (2016) performed experiments with freshwater addition larger than estimates (up to 3000 Gt yr-1 increase) with a fully coupled model. Using two different freshwater distributions (ice shelf melt in front of ice shelves and at the depth of the front, and iceberg melt at the surface over a wide area), they found that the total sea ice area increases significantly under the large freshwater enhancement. In accordance with the similarities we see between our experiments S2-High and S3-High after 10 years of integration, their experiments show that the spatial distributions of the freshwater input have no significant influence on the sea ice response. However, S2-High and S3-High feature divergent behaviour on seasonal time scales. As Pauling et al. (2016) point out, differences in the model complexity (as forced vs. fully coupled configurations) and in their physics could lead to diverging results. With the low stability of the Southern Ocean water column, small differences in the chosen parameterizations, e.g. in the vertical mixing, can have a large effect on the sea ice. This is especially known for the Weddell Sea, which is the main region of sea ice loss in our experiments."

In Section 5 (formerly 4), we removed the paragraphs referring to S5.

In the concluding Section 6 (formerly 5; p. 15, l. 10):

"However, we find a strong dependency on the amount of freshwater addition, as in our model large amounts of fresh water can affect the sea ice trend inversely. Our experiments with the strongest freshwater forcing result in a decrease in sea ice extent and volume."

3. In relation to this, please can you explain why the simulations of sea ice are considered to represent sea ice behaviour, while the simulation period of 10 years is too short for the water characteristics to

reach equilibrium (see e.g. p. 2, line 28-33). Are you saying that you are investigating sea ice response processes and therefore do not need to reach equilibrium? If this is the case, I am not sure I understand how you may conclude that there is a reversal of behaviour when more than a certain amount of fresh water (undetermined from these experiments) is added to the system. How can you tell that this is not due to variability between runs? This may require more explanation of the known behaviour of the model. The existence of a turning point based on evidence of a single simulation requires additional argument for its existence.

We performed our sensitivity experiments with the aim to analyse the very-short-term response of the sea ice and upper ocean to an idealized "perturbation" of the system. The 10 years of integration are not sufficient to let the ocean reach equilibrium after the disturbance at the surface. Comparing the model results to the reference simulation within this time window, we investigate the sea ice behaviour and the feedbacks with the water masses (mainly on the continental shelf areas) under an imposed "climate change" (as may currently be observed in the real sea ice/ocean), not the state the system would have reached after re-finding its balance. For this, we are more interested in the qualitative response and the underlying processes than a quantitative assessment and an exact reproduction of observations.

We compare experiments, which differ only in the addition of freshwater. The different results are reliably caused by the differences in runoff input.

We agree with the referee that a stronger corroboration of the mentioned turning point by additional experiments is beneficial. We performed two additional experiments with even higher freshwater addition, which both resulted in sea ice loss, thus substantiating the possible reversal of the sea ice trend (at least in our model). (See also answer to Main comments #2)

In Section 2.2, we added the following text: "The sensitivity experiments are run from 2004 to 2013. The 10-year period is not sufficient to let the 3D ocean reach equilibrium after the disturbance at the surface. However, this study only aims to examine the very-short-term response of the sea ice and upper ocean system to the imposed idealized changes in the Southern Ocean freshening, via the comparison to the CTR run. "

4. How was the seasonal variation in ice shelf "runoff" decided (see Fig 1e)?

There is not much known about the seasonality of Antarctic runoff. However seasonality can be expected for both, iceberg meltwater and basal melting. In the first case the seasonality is strong (e.g. Merino et al., 2016) since the ocean surface heats up in summer. In the latter case uncertainties are large, but in winter the dense water formation in coastal polynyas inhibits warm water intrusions under the ice shelves and therefore a higher heat flux into the cavities can be expected during the summer.

The runoff applied in our reference run (CTR) is derived from the river runoff dataset by Dai and Trenberth (2002) and Dai et al. (2009) for the major rivers, while the Antarctic coastal freshwater discharge is based on estimates by Jacobs et al. (1992). Bourdalle-Badie and Treguier (2006), who joint the two datasets and adapted them to the ORCA025 model grid, also applied a *pseudo* annual cycle to the Antarctic contribution, with the maximum of ~0.15 Sv in December/January and the minimum of 0.04 Sv in June/August.

Bourdalle-Badie, R. and Treguier, A.-M.: A climatology of runoff for the global ocean-ice model ORCA025, Report, MercatorOcean, MOO-RP-425-365-MER, 2006.

Dai, A., T. Qian, K. E. Trenberth, and J. D Milliman, 2009: Changes in continental freshwater discharge from 1948-2004. J. Climate, **22**, 2773-2791.

Dai, A., and K. E. Trenberth, 2002: Estimates of freshwater discharge from continents: Latitudinal and seasonal variations. J. Hydrometeorol., **3**, 660-687.

Jacobs S. S., H. H. Hellmer, C. S. M. Doake, A. Jenkins, R. M. Frolich (1992), Melting of ice shelves and the mass balance of Antarctica, J. Glacio. 38, 375-387.

5. Development in time and variability on p. 9: How much is know about variability between model runs when there has not been a repeat of an experiment? Perhaps this is well known for the model and could be briefly explained to the reader.

The model variability is low and a repeat of any experiment is expected to give the same results, since the runtime of only ten years does not give the small numerical errors the time to grow into variability of any significance.

In the article, we refer to the variability of the change effected in a sea ice property, i.e. whether a (detrended) time series has a high or low standard deviation.

6. Comments 2-5 lead me to be unconvinced by the authors' conclusion that (the small) freshwater input they apply causes the sea ice to expand, while a larger input inverts the trend. This needs to be very carefully re-evaluated.

We conducted further experiments to substantiate our results (see Main comments #2 and #3). We would like to mention here, that there are no changes in the initial conditions in our experiments or any other introductions of randomness. We do not consider the results of a singular experiment. We always compare two experiments that only differ in the freshwater input. The differences in output between our experiments are therefore not random, but the result of the changes introduced by the modifications of the freshwater.

**TECHNICAL COMMENTS**

p. 2, line 9-10: Merino et al and Pauling et al (2016) need to be added to the previous studies.

As suggested, we included the studies by Merino et al. (2016) and Pauling et al. (2016) in the revisited manuscript. We added the following text in Section 1 (p. 2, l. 32 - p. 3, l. 5): "Merino et al. (2016) used an iceberg model coupled to a sea ice-ocean model to establish a seasonal climatology of iceberg melt for the Southern Ocean. They find that the iceberg melt water leads to higher sea ice concentration and thickness, with exception of the Amundsen/Bellingshausen Sea area. Pauling et al. (2016) employed an Earth-system model to investigate the effect of artificially augmented, constant freshwater input on sea ice. They tested the sensibility to freshwater additions from current estimates to much larger values (3000 Gt yr-1) and compared an iceberg model-based surface distribution with a coastal distribution at the depth of the ice shelf front. Their model experiments show that enhancing the freshwater input by an amount within the range of estimates of the Antarctic mass imbalance does not significantly affect sea ice area, magnitude or trend. Further, their results show the sea ice response to be insensitive to the depth of fresh water injection."

We discuss the consistency of the sea ice response in our sensitivity runs and experiments by Pauling et al. (2016) in Section 4 (p. 13, l. 8-18): "Our results with "extreme" freshwater input suggest that sea ice trend is sensitive to the amount of freshwater and to the method by which it is added. Pauling et al. (2016) performed experiments with freshwater addition larger than estimates (up to 3000 Gt yr-1 increase) with a fully coupled model. Using two different freshwater distributions (ice shelf melt in front

of ice shelves and at the depth of the front, and iceberg melt at the surface over a wide area), they found that the total sea ice area increases significantly under the large freshwater enhancement. In accordance with the similarities we see between our experiments S2-High and S3-High after 10 years of integration, their experiments show that the spatial distributions of the freshwater input have no significant influence on the sea ice response. However, S2-High and S3-High feature divergent behaviour on seasonal time scales. As they point out, differences in the model complexity (as forced vs. fully coupled configurations) and in their physics could lead to diverging results. With the low stability of the Southern Ocean water column, small differences in the chosen parameterizations, e.g. in the vertical mixing, can have a large effect on the sea ice. This is especially known for the Weddell Sea, which is the main region of sea ice loss in our experiments. "

p. 2, line 24-25: Note that Pauling et al (2016) have added fresh water spatially distributed according to ice shelves, and at the depth of the ice shelf. However their simulations did not vary in magnitude through the year.

The fact that their fresh water addition was constant is now mentioned in the text. In the Introduction (Section 1, p. 2, l. 34), we say: "Pauling et al. (2016) employed an Earth-system

model to investigate the effect of artificially augmented, constant freshwater input on sea ice."

p. 2, line 28-33: (as main comment) Please can you explain why the simulations of sea ice are considered to represent sea ice behaviour, while the simulation period of 10 years is too short for the water characteristics to reach equilibrium. Are you saying that you are investigating sea ice response processes and therefore do not need to reach equilibrium? If this is the case, I am not sure I understand how you may conclude that there is a reversal of behaviour when more than a certain amount of fresh water (undetermined from these experiments) is added to the system. How can you tell that this is not due to variability between runs? This may require more explanation of the known behaviour of the model. The existence of a turning point based on evidence of a single simulation requires additional argument for its existence.

Answered above (Main comments #2 and #3)

p. 4, line 5-6: Was Dai and Trenberth (2002) applied in all other parts of the globe, apart from Antarctica? Was the seasonal variation used (see Fig 1a – actually I think it is 1e) from Dai and Trenberth (2002)? If so how do you justify using the seasonal behaviour for river runoff to represent melting ice shelves?

For most of the global domain the river runoff is a monthly climatology derived from the freshwater discharge dataset by Dai and Trenberth (2002) and Dai et al. (2009). Their dataset includes the river outflow for the largest 925 ocean-reaching rivers on global scale. The following plot (Figure 1 in Dai et al. 2009) shows the locations of the gauge stations used in their study. Unfortunately, their estimate of the global freshwater discharge excludes an estimate of the freshwater flux from Antarctica, as reported in Section 4. in their manuscript "…our estimates of the continental discharge include runoff from all land areas except Antarctica..." The contribution of Antarctica freshwater applied in our reference simulation relies on estimates of 2613 kg y-1 (0.0829 Sv) by Jacobs et. al. (1992). Bourdalle-Badie and Treguier (2006) distributed the Antarctic freshwater contribution as a uniform flux along the coastal points around the Antarctic continent and applied a *pseudo* annual cycle to it (as described in our answer to comment #4).

To more clearly explain this, we changed the description of the runoff data set in Section 2. as follow: "The runoff data is a monthly climatology derived from the Global River Flow and Continental Discharge Data Set (Dai and Trenberth, 2002; Dai et al., 2009) for the major rivers and estimates by Jacobs et al. (1992) for the Antarctic coastal freshwater discharge. It has been adapted to the ORCA025 grid and applied along the land mask (Bourdalle-Badie and Treguier, 2006)." We added the references. The melt of the Antarctic glacial ice in the Southern Ocean is primarily dependent on the water temperature. For the basal melt of the ice shelves the main question is therefore how much warm water can intrude onto the continental shelves. In winter, deep convection linked to polynya activity hinders the warm water intrusions and thus higher melt rates can be expected in the summer months. Also for iceberg melt the seasonal dependence is strong due to the surface warming (Merino et al., 2016). We admit that there is limited knowledge of the seasonal cycle of the Antarctic 'runoff'. The runoff in our study therefore may not be correct in amplitude or shape, but some seasonal variation of the meltwater may be expected.

p. 4: Table 1 is very useful but has not been referred to in the text. It would be useful to refer to it in section 2.2.

We added the following sentence in Section 2.2 (p. 5, 1. 11). "A short overview of the experiments and their differences is also given in Table 1."

p. 4, lines 12-33: I think that the subfigures of Fig. 1 have been mislabeled. Yes, we apologize for the mix-up. The mistake is now corrected.

p. 4-5: Experiment design – please note that Merino et al (2016) and Pauling et al (2016) both conduct experiments with fresh water distributed to mimic iceberg melt.

In this section, we describe only our own experiments. However, both suggested studies are now added with mention of the iceberg model-derived distribution in the Introduction (Section 1.) and our findings are compared with their results in Section 4 (formerly 3.3). (See also Main comment #1).

p. 5, line 20 onwards: This is a very interesting discussion regarding the influence of additional fresh water at the surface on the SSH, the velocity and thus on sea ice thickness. I was confused about how changes in the direction of the velocity were taken into account? Does the right hand column of Fig 2 show speed not velocity?

The right column of Figure 2 shows the differences in sea ice velocity between a single sensitivity run and CTR as arrows and the corresponding differences in vector magnitude (speed) as filled coloured contours. We modified the figure caption as follows:

(p. 21) "Figure 2: Maps of winter sea ice a) concentration, b) thickness, and c) velocity in CTR averaged over April - September 2004 – 2013. b-r) Difference of ice concentration (left), thickness (middle), and velocity (right) between respective experiment and CTR. The colors underlying the velocity arrows indicate the difference in vector magnitude (speed). Dark red contours encompass the areas where the significance of the difference surpasses the 99% confidence-level of the Student t-test for dependent samples."

p. 5: Spatial Response Patterns: How can you have a high confidence interval in the difference when, at each time step, there are only two quantities? Is it time-averaged?

The confidence level was determined using the Student's t-test for dependent data samples. The 'sampling period' was limited to the 10\*6 monthly means April-September 2004-2013.

Fig 2 is for the "winter" months. Which months are "winter"?

We do refer to 'winter' as the 6-month period from April to September as now clearly stated in Section 3, p. 6, l. 20: "*In the following, the word winter referring to a specific time period identifies the period April – September.*"

p. 6: line 9-10: Is a salinity-dependent freezing point coded in the model? Yes, in the LIM2 sea ice model, the freezing temperature of seawater depends on salinity, linearly with an empirical constant (Vancoppenolle et al., 2012).

Vancoppenolle, M., Bouillon, S., Fichefet, T., Goosse, H., Lecomte, O., Morales Maqueda, M. A., and Madec, G.: The Louvain-laNeuve sea ice model, Notes du pole de modélisation, Institut Pierre-Simon Laplace (IPSL), Paris, France, no. 31, ISSN No 1288-1619, 2012.

p. 6, line 23 + p.7, lines 14, 27, 28, + p. 8, line 12, + p. 13 line 23: use of the word "acceleration" when I think you mean "faster speed"

Yes, not in all cases the words accelerated/acceleration were used in their proper sense. We corrected the phrasing where necessary.

p. 7, line 26: please mark Princess Martha Coast on a map. Princess Martha Coast was marked on the map in Fig. 1b)

p. 8, line 10: please mark Filchner/Ronne Ice Shelf on a map

We consider the Filchner/Ronne Ice Shelf to be a well known feature of the Antarctic geography like e.g. the Ross Ice Shelf, Amundsen Sea and Antarctic Peninsula. We are afraid that we cannot mark all the place names mentioned in the manuscript within our Figure 1. We therefore kindly ask readers unfamiliar with the main features of the Antarctic geography to refer to a map from another source.

p. 9, line 10-12: Why are the larger amplitude anomalies in 2009-2011? Why are the anomalies smaller in 2012-2013?

We cannot answer this question completely. As mentioned in the article, regional time series show that the difference occurs in the Amundsen, Bellingshausen and western Weddell Seas. We therefore assume that the atmospheric circulation features a regional shift in those years that has a much stronger effect on the experiments with regionally varied runoff than on the experiments with uniform coastal runoff.

p. 10, line 9-15: We are not shown the surface salinity or the SST so it is difficult to follow this discussion. Could the essential elements be presented in a figure?

Since the question of the experiment S5 (now S2-Incr) has in the meantime changed character and the manuscript has been adapted to include the new runs, we decided not to include the figure in the article. However, we are happy to include figures of the differences in SSS and SST between the runs S2-High and S2, and S3-High and S3 here in our response (Figure R1). We hope that this serves as helpful illustration beyond the scope of the article itself.

---

## Author Response (AR3)

**Authors' response to Referees' comments**

We would like to express our gratitude toward our reviewers and our editor for many helpful suggestions and corrections of our manuscript. We hope that all remaining questions are addressed and answered in a satisfactory manner in this response.

Submitted on 14 Mar 2017
Anonymous Referee #2

**Suggestions for revision or reasons for rejection (will be published if the paper is accepted for final publication)**

The revisions made to this paper are a substantial improvement and I congratulate the authors on a nice piece of work. The authors have carried out additional simulations that confirm the response of their model to "large" amounts of freshwater. I have two remaining queries about the manuscript.

Comments

Section 3.1 & Figure 2: I am still not clear what is meant by variability and what dependent samples are being selected.
We agree. The use of the word "variability" in this section is unclear. We actually refer to the changes in time of the differences in ice concentration or ice thickness. We changed the text from *'The variability of the differences …'* (p. 7, l. 4 in the previous version) to *'The changeability in time of the ice concentration differences between experiments S1-S4 and CTR is comparatively high, …'* (p. 6, l. 29), and from *'… reduces the variability…'* (p. 7, l. 8 in the previous version) to *'… reduces also the changeability of the ice thickness differences in time...'* (p. 7, l. 2).
In the Student's t test, the samples are the relevant data points of the time series (winter months) from each experiment respectively. The two samples (one from CTR and one from a sensitivity experiment) are dependent, since the sampling is identical in time and the only source of the differences is the investigated perturbation (here: the applied modification in the freshwater input). For no change in freshwater input, identical results are expected. The test therefore sets the mean of the differences in relation to the standard deviation of the differences in order to determine whether the mean is a reliable measure of the differences (statistically significant) or a product of chance.
With the sample size N and the two paired samples A and B, the value of T is calculated as follows:
T=sqrt(N)*abs(mean(A-B)/stddev(A-B))
The critical value for N=60 and 99 % confidence is T=2.618.

I feel that some of the Conclusions (in particular, p. 15, line 1-6) are not supported by evidence given in the paper. Please clarify the source of this information.
We see how some formulations could be misunderstood or might overshoot the mark (see also the list of technical corrections below). Specifically, the remark on the increased stratification was too general and, unfortunately, we had a mix-up with the SST.
Since the possible influence of freshwater distribution cannot be denied concerning S2-Incr, S2-High and S3-High, we removed the statement on dependence on the amount of freshwater preferring to err on the side of caution.
The fist sentence of the Conclusions was slightly modified: *'We have investigated the hypothesis that increasing freshening of the Southern Ocean could explain, at least partially, the Antarctic sea ice*

*expansion.'* (p. 14, l. 9-10)
The second paragraph in the Conclusions section has been modified and now reads as follows:
*'Our results confirm that the sea ice extent (and volume) increases for moderate increases of the runoff amount. As also indicated in previous studies (e.g., Swingedouw et al. 2008, Bintanja et al, 2013), the enhanced freshwater input increases the near-surface stratification, which, in turn, inhibits the vertical transport of heat from depth to the ocean surface, a situation that can foster the formation of sea ice. However, we also find that, in our forced configuration, a large amount of freshwater can affect the sea ice trend inversely. Our experiments with the strongest freshwater forcing result in a decrease in sea ice extent and volume.'* (p. 14, l. 14-19)

We also revised the fourth paragraph in the Conclusions section:
*'Our model results show that the addition of freshwater can induce a warming in the sub-surface waters where the halocline and the near-surface stratification are strengthened and the vertical heat exchange is reduced. On the Antarctic continental shelves, the water characteristics are therefore subject to significant changes. In our experiments, the immediate and possibly transient response of the dense shelf water characteristics is a warming and freshening for simple increases in the runoff. The dense shelf waters hence become less dense. However, in regions downstream of reduced freshwater input at the coast, the water column is less stable and in consequence waters generated on the shelf are denser (colder and more saline).'* (p. 14, l. 25-30)

Technical corrections

Suggest removing "the" from title
Following the suggestion, the title now reads: *'Impacts of freshwater changes on Antarctic sea ice in an eddy-permitting sea ice-ocean model'* (p. 1, l. 1-2)

p. 2, line 26: change "in" to "is"
Sorry for this oversight. We corrected it. (p. 2, l. 29)

p. 3, line 1: replace "sensibility" with "sensitivity"
The change was done as suggested: *'They tested the sensitivity to freshwater additions...'* (p. 3, l. 4)

p. 3, line 6-8: "Our study aims to investigate the sensitivity of sea ice properties including the dynamic response to changes in the amount and especially the spatial distribution of fresh water input at surface including also the dynamic response of the sea ice-ocean system." is poor English and the meaning is not clear.
To improve readability and make it easier to understand, we rephrased the text as follows: *'Our study investigates the short-term response of sea ice and upper-ocean to prescribed changes of the surface freshwater input, both in magnitude and spatial distribution. We do consider these modifications as a perturbation of the current-climate environment. We specifically include the dynamic response of the sea ice-ocean system in our analysis.'* (p. 3, l. 9-11)

p. 3, line 12: Insert "the" in "at the surface"
Changed to: *'… Antarctic surface freshwater flux...'* (p. 3, l. 15)

p. 3, line 20: "insight to possible"
The sentence was changed as follows: *'… allow us to gain insight into possible future sea ice changes.'* (p. 3, l. 23-24)

p. 5, line 29-30: Replace "… are associated to the iceberg freshwater release (Rignot et al., 2013), and are widely distributed (with four levels of flux intensity) in the Southern Ocean (with four levels of flux intensity)" with "… is associated with the iceberg freshwater release (Rignot et al., 2013), and is widely distributed (with four levels of flux intensity) in the Southern Ocean (with four levels of flux intensity)"

Done as suggested: *'… while 1090 Gt yr$^{-1}$ is associated to the iceberg freshwater release (Rignot et al., 2013), and is widely distributed in the Southern Ocean...'* (p. 5, l. 32-33)

p. 7, Section 3.1: I am still not clear what is meant by variability and what dependent samples are being selected in Figure 2.

Already answered (please see above the answer to the first comment).

p. 8, 3-4: "The sea ice … coastal area." is not a sentence.

To increase readability, we modified the sentence: *'For the sea ice concentration (Figure 2g), we find changes of high statistical confidence in the coastal area.'* (p. 7, l. 30-31)

p. 8, line 31: Do you mean "acceleration"? Do you not mean "greater speed"? What do you mean by an "inverted" acceleration?

We agree that our choice of words may lead to misunderstandings. With 'acceleration' we actually referred to an increasing, positive difference in speed and by 'inverted' acceleration, we refer to a sign change (subsequent to a decrease) of the speed difference between two experiments. To increase the clearness of our manuscript *'… the strong acceleration experienced in the Amundsen and Ross Seas, is inverted while circling East Antarctica…'* (p. 8, l. 31 in the previous version) was replaced by: *'… although in the Amundsen and Ross Seas the current increasingly gains speed compared to CTR, while circling East Antarctica the differences in speed between S2 and CTR decrease and eventually change sign...'* (p. 8, l. 26-27)

p. 11, line 9: spelling "experiments"
The mistake was corrected.

p. 11, line 10: Replace "similar magnitude as" with "similar magnitude to"
The sentence was modified: *'… and in magnitude comparable to…'* (p. 11, l. 4)

p. 11 line 29 – p. 12, line 10: I do not understand why a curve is fitted to the difference between S1 (or S4) and CTR and projected to 35 years, rather than running the model for 35 years. Please could this be rewritten to explain.

We used the projection instead of running the experiments for a longer period to limit the computational cost. We do agree with the Referee that the fitting of a curve to the differences and subsequent projection is a source of errors. The comparison is flawed by more influential assumptions, as it is an attempt of comparing a trend caused by slow changes (as it occurs in the observations) with differences caused by an instantaneous change (as in the model). We have therefore decided to remove the comparison with the observed trend.

p. 12, line 12: replace "like" with "as found in simulations by"
Changed as suggested.

p. 12, line 16: Suggest changing "as also Merino et al. (2016) found, considering an" to " we confirm the results of Merino et al. (2016) who showed an"
According to the suggestion we changed the sentence to: *'In particular, our results confirm the results*

*of Merino et al. (2016) who showed that an idealized...'* (p. 11, l. 26)

p. 12, line 21 onwards: This section of text does not make sense in relation to Figure 5. I suspect you mean Figure 4?
We apologize for the mix-up. We now corrected the references to Figures 4 and 5:
*'… sea ice starts to decrease (Figure 4). '* (p. 12, l. 6)
*'The seasonal mean (Figure 4) still bears …'* (p. 12, l. 8)
*'… computed in all three regions (Figure 5a-c, g-i, m-o), …'* (p. 13, l. 12)
*'… the simulated decade in Figure 5(d-f, j-l, p-r).'* (p. 13, l. 19)

p. 12, line 23: Suggest replacing "objective to verify and understand" with "objective of verifying and understanding"
Correction done as suggested. (p. 11, l. 33-34)

p. 13, line 15: remove plural i.e. "ice sheet and iceberg basal"
Changed as suggested: *'… fresh water from ice sheet and iceberg basal melt…'* (p. 12, l. 25)

p. 13, line 28: The figure labels do not correspond with the description.
The figure reference was corrected: *'… computed in all three regions (Figure 5a-c, g-i, m-o), …'* (p. 13, l. 12)

p. 14, line 4: The figure labels do not correspond with the description.
The figure reference was corrected: *'… the simulated decade in Figure 5(d-f, j-l, p-r).'* (p. 13, l. 19)

p. 15, line 2: I do not recall any place in the paper where increased ocean stratification was demonstrated – it is not shown in the Figures. Similarly there are no plots of SSTs. If these are in the supleementary material please refer the reader there. If it is not shown but seen in the model please comment. If it is a hypothesis please state that clearly.
We agree with the Referee that the stratification is not necessarily increased by the freshwater addition over the whole water column and that our statement was too rigorous.
The initial increase in the near-surface stabilization that limits the vertical transport of heat to the ocean surface can cause a sub-surface warming. The stabilizing effect of the fresher surface water is therefore counteracted by the warming. Figure 5 shows that, in most cases, the surface freshening is accompanied by a sub-surface warming. Since Figure 5 only shows means over large areas of the shelves, we included in our response here (Figure R.1) two exemplary sections, that show this behaviour. In the first case (Figure R.1a-f), the stability is increased only in the near surface areas where the salinity gradient is dominant. In the second case (Figure R.1g-l), although there is a warming of the sub-surface waters, the stability of practically the entire water column on the shelf is enhanced. We agree with the referee that the statement on SST in direct relation with sea ice formation was faulty. Where sea ice formation takes place, we expect SST to rise. The minimum of SST is coupled to the freezing point of sea water, which is higher for less saline water. We changed our formulation of the sentence: *'As also indicated in previous studies (e.g., Swingedouw et al. 2008, Bintanja et al, 2013), the enhanced freshwater input increases the near-surface stratification, which, in turn, inhibits the vertical transport of heat from depth to the ocean surface, a situation that can foster the formation of sea ice.'* (p. 14, l. 14-17)
Additionaly, we modified a sentence in Section 3.1.1: *'The surface freshening by the enhanced runoff reduces the density of the surface layer and therefore leads to a strengthening of the ocean stratification (not shown), ...'* (p. 7, l. 20-21)

[Figure]

Figure R.1: Sections of mean salinity, temperature and $\log(N^2)$ in the a-f) Ross Sea (177.5°/-77° to -175°/-74.6°) and g-l) Weddell Sea (-61.3°/-68.3° to -52.5°/-68.3°) for the period April-September 2004-2013. The CTR values are shown above a difference plot.

p.15, line 26-27: Pauling et al. (2016) have added water at non-zero depth. This sentence implies it has not been domne before. I am not sure what you mean by this sentence, but suggest you either qualify that it refers to simulations in NEMO, or delete the sentence.
Although, we intended it as a caveat of our study and never wanted to imply it has not been done before. We modify the text as follows: '*It is worth noting that the impact on local ocean and sea ice, simulated in our experiments, is due to freshwater input that enters the ocean only through the surface. The sea ice response and the consequent impact on the water mass characteristics reproduced by our model may be sensitive to the depth of freshwater injection, and possibly be improved by better representation of the calving and basal melting of the ice shelves.*' (p. 15, l. 5-8)

Fig 2: I think the caption should say "d-o)" rather than "b-r)". As mentioned above, I am still confused regarding how the variability was defined to carry out the Student t-test.
Yes, the caption was wrong. We changed it to '*d-o) Difference of…*' (p. 21, l. 4) and hope our answer to your comments (above) helps clear up the issue of how the Students t-test was performed. We also added a short explanation to the caption of Figure 2: '*... the Student t-test for dependent samples, where the mean difference between two samples (here: time series) is set in relation with the standard deviation of the differences.*' (p. 21, l. 7)

I think the labels of Fig 4 and Fig 5 have been mixed up.
Yes, we corrected the references in the text.

Fig 5 (that should be 4): The labeling of the sub-figures does not correspond with the text.
We thank the Referee for pointing that out. The order of sub-figures has been corrected in the Figure.

Submitted on 24 Feb 2017
Anonymous Referee #3

**Suggestions for revision or reasons for rejection (will be published if the paper is accepted for final publication)**

General comments:
In this manuscript Haid and colleagues investigate the impact of freshwater from Antarctica on surrounding sea ice. They vary both the regional distribution and the magnitude of freshwater released at the surface of the ocean. I think the subject is very important to understand the past sea ice trend and to improve future projections. The contradictions in the literature about the ability of the freshwater trend to be the cause of the sea ice trend of the past decades make it even more timely to investigate. The eddy-permitting model used in this study and the variety of experiments performed gives a lot of potential to this study. Unfortunately the manuscript is written like a description of all the work that has been performed instead of as a well focussed demonstration of the main points. The editor's comment still holds for this version of the manuscript, I advise the authors to sharpen their arguments, possibly focus on a specific point and ideally perform longer simulations. Some suggestions are detailed below.

We are grateful to the reviewer for his/her in-depth revision of our work. We have followed the suggestions to add comments and support the arguments that we made with new analyses. The set of experiments is however the same and their lengths have not been increased for reasons that we explain below. We hope that the additional figures that we provide in the following answers can support and better explain our finding.

Probably due to the fact that the eddy-permitting model used is expensive to run, the experiments are very short with large inter-annual variability and still do not seem to be close to adjustment at the end of the 10 years. As the editor and reviewer #2 I think this is an important issue. For this reason, I do not think that it is reasonable to extrapolate these short time series to give a range of the possible contribution of freshwater trend to sea ice trend over the last 35 years. There is already enough contradiction in the literature. To tackle this problem a 35 years simulation should be done using the best freshwater distribution, probably S3 type and a realisticaly increasing freshwater forcing. Furthermore in making this exercice of relating the experiments S1 and S4 to real world trend two assumptions have been made: (1) Impact of freshwater on sea ice scales linearly with the amount of freshwater and (2) it is not dependent on regional pattern. These are exactly two assumptions the authors argue are not valid based on other experiments.
We are sorry that the purpose of our analysis on the contribution of freshwater to sea ice trend projected over 35 years was not clear. Due to the *idealized* nature of our experiments, a direct comparison with observed data and observed trend is beyond the scope of this study. The Referee is correct in pointing out that the comparison with the observed trend is based on poorly founded assumptions. We agree that the benefit of the calculation to our study is rather small and, to avoid misunderstanding, we have decided to remove the comparison to observed trends from the manuscript.

I find the experiments S1, S2 and S3 interesting because the pattern of freshwater forcing is more and more realistic with almost the same amount of total freshwater (why not exactly the same?). The demonstration that regional pattern is important is convincing but better analysis of the mechanisms is necessary.

We aimed to examine the sensitivity of sea ice response to both magnitude and distribution of the surface freshwater forcing. We modified the spatial distribution of the freshwater input, from one sensitivity experiment to another, toward a more and more realistic representation of the complex real distribution of ice shelf and iceberg meltwater. At the same time, we investigated the effect of a direct addition of freshwater by using a set of increasing values, overestimating the observed estimates. Our study focuses on the qualitative (rather than quantitative) response of the sea-ice/ocean system to runoff *perturbation*. We do not consider it very relevant, for the purpose of the study, if the amount of additional fresh water in S1 exactly matches that of S2 and S3.

As to an improved description of the processes, we modified the related text in Section 3.1, as follows: '*Sea ice velocity is also affected by the changes in runoff. The addition of fresh water locally increases the sea surface height (SSH) in two ways: 1) directly by the addition of the fresh water's volume to the ocean and 2) by decreasing the density of the upper water column. This affects both the SSH slope and the density gradient, which influences the surface current of the ocean and the sea ice drift. Changes in the sea ice velocity influence the sea ice thickness. By an increased drift velocity, the time available for thermodynamic growth can be shortened, but in zones of sea ice convergence the thickness is increased through the dynamic compaction processes such as rifting and rafting.*' (p. 7, l. 4-9)

It is not made clear how the freshwater leads to changes of the current. The reason given by the authors is: "The addition of fresh water affects the sea surface height (SSH). A change in the SSH slope influences the surface current of the ocean and the sea ice drift". I assume what is meant is that freshwater leads to decrease density close to the coast, which produces higher SSH, and that this strengthen the meridional density gradient and westward surface currents as a results of thermal wind balance. Is this the mechanism?

The changes in ocean currents and related convergence of the sea ice explain some of the differences between experiments. Since it seems to be an important mechanism it should be computed and shown on a map. This way it is easier to explain the changes in sea ice thickness.

The mechanism that we think is in play in our simulations corresponds to the description suggested by the Referee, although a further effect on the SSH is the plain addition of water volume. To account for this comment, we changed the cited sentences to the following: '*The addition of fresh water locally increases the sea surface height (SSH) in two ways: 1) directly by the addition of the fresh water's volume to the ocean and 2) by decreasing the density of the upper water column. This affects both the SSH slope and the density gradient, which influences the surface current of the ocean and the sea ice drift.*' (p. 7, l. 4-7)

In Figure 2, we show maps of the changes in sea ice drift velocity (and thickness). Table 2 gives evidence that the thermodynamic response of the sea ice is not the full story. We think that Figure 2 and Table 2 clearly show that sea ice convergence is a major factor influencing the response of the sea ice thickness. We thank the referee for the suggestion, but we do not consider that adding a new figure to show the sea surface height (SSH) (while truly an important part of the puzzle) is fundamental for the manuscript since cause (freshwater input) and effect (ice velocity) are presented.

To properly answer to the Referee's comment, we add Figure R.2 to this response. The plot shows the SSH in CTR and the comparisons with S1, S2 and S4, and highlights the link between differences in freshwater flux and the consequent changes in SSH. An extra freshwater input generally results in increased SSH and SSH slope. The influence of advection of the fresh water out of the immediate areas of addition is visible pariculary in Figure R.2d). Similarly, in Figure R.2c) the advection of the Ross Sea waters causes the differences between S2 and CTR to slowly decrease along the coastline of East

Antarctica. In S2, the reduction of the SSH in the Bellingshausen Sea and the Eastern Weddell Sea coincides with the reduction of freshwater input (in those same areas) compared to CTR. We also add Figure R.3 here, that presents the divergence of the sea ice velocity in CTR and the comparisons with S1, S2 and S4. Despite the high spatial variability of the presented quantity, for all three experiments we can see patches of increased convergence in the western Ross Sea and along the Antarctic Peninsula in the western Weddell Sea.

We do not consider that the maps can largely improve the explanation of the mechanism of ice thickness increase due to deformation. Convergence in a very small area can have a strong effect on the thickness, while divergence in large areas has practically no influence on the ice thickness. Also, when looking at differences, the total is very important in this case. Where divergent ice flow is found, a lessening of the divergence (neg. difference) is of no influence, while in an area of convergent flow the same difference is of great impact. We think that the main features can also be seen in the maps of ice velocity in Figure 2 in the manuscript.

[Figure]

Figure R.2: Maps of the mean sea surface height a) in CTR and b-d) differences between respective experiment and CTR averaged over April - September 2004 – 2013.

[Figure]

Figure R.3: Maps of the divergence of the sea ice velocity a) in CTR and b-d) differences between respective experiment and CTR averaged over April - September 2004 – 2013.

I feel that the finding on the potential decrease sea ice due to high amount of freshwater is very important but the demonstration is not very convincing. First the argument that a small amount of freshwater increases sea ice volume can only be made by comparing CTR and S1 since for S2, S3 and S4 the pattern is also changed which might have more impact than the amount. Then the argument that large amount of freshwater reduces sea ice is made with S2 and S3 experiments. This is not consistent. Also this is such a large disturbance to the system that I would expect the adjustment to take more time in the "High" simulations than in S2 and S3, which are already not equilibrated. Could this just be a peculiar transient adjustment?

We find that the argument that the addition of a small amount of freshwater increases sea ice volume can be made with S4 just as well as with S1, since both only feature increases in freshwater addition compared to CTR. We do not expect any increases of Antarctic meltwater to result in a uniform distribution around the coastline and should therefore not restrict our investigations to such cases.
Since the argument was made without any restrictions – additional freshwater **can** lead to increases in sea ice – we also include the results from S2 and S3, although (as the Referee points out) in those runs, in addition to the increase of fresh water, there is also the shifts of freshwater from one location to another.
Yes, we agree with the Referee, the 10-year simulation period is not sufficient to let the system reach

equilibrium after the disturbance at the surface. We do consider the modification of the freshwater input as a perturbation of the current-climate environment. Our objective is to analyze the very-short-term response of the sea-ice/upper-ocean system to the imposed idealized *climate changes,* via the comparison to the control run, (as may currently be observed in the real sea ice/ocean), not the state it will reach after re-finding its balance. The reduced sea ice in "High" simulations may be a transient adjustment, but in all probability so are the trends recently observed in the Antarctic sea ice. We investigate the immediate response of the sea ice in order to gain some insight into a major factor that potentially affects the trends we observe in the ocean/sea ice system. The modelled system is in a transience state, not in equilibrium. We therefore refrain from any absolute quantitative statements and focus on the direction (or the sign) of the response and the magnitude of changes relative to each other. Primarily, we want to identify connections and the underlying mechanisms.

To underline our interest in the immediate response of sea ice and ocean and rely it more immediately the the reader, we added one word to our Abstract:

*'We analyse the short-term response of the sea ice...'* (p. 1, l. 13)

As the authors say, the impact of freshwater on the dense shelf water are quite complex because it seems to depend on the regional distribution of the freshwater. I do not see a connection between this part of the manuscript and sea ice. I am sure there are some feedbacks between dense water formation and sea ice, probably also with time scales longer than 10 years. I think these results should be added in a paper focussed on see ice only if a clear connection is made between the two.

We agree with the Referee that the link between on-shelf water column and the sea ice was not clearly assessed. We have now added an introductory paragraph in Section 5, which tries to fulfil this purpose.

*'Sea ice and ocean are two components of the Earth's climate system that are strongly linked. Accounting for their complex interplay is an important part of a precise reproduction and prediction of the climate. Sea ice formation and melt, which are strongly influenced by the availability of oceanic heat at the surface, directly affect the properties of the near-surface water masses. Sea surface salinity (SSS) can either be increased due to brine exclusion (during the formation of sea ice) or decreased due to sea ice melting. Sea ice formation and brine exclusion rates play a vital role in the southern circulation regime and largely control the formation of the dense waters.*
*In this section, we present the effect of the freshwater additions in our experiments S1-S4 on the water columns in the key regions for dense water formation: the continental shelves of the Weddell Sea, the Ross Sea and the Prydz Bay.'* (p. 12, l. 31 – p. 13, l. 4)

The use of an eddy-permitting model is an important novelty but it is not justified and analysed enough. Why is higher resolution expected to be better for this problem? Probably the coastal current described by the authors is better represented but is it stronger or weaker than in a low resolution model? These results could be compared with a low resolution configuration of NEMO to answer these questions.

As the Referee mentions, higher resolution in a model allows for a better representation of features that take place on small spatial scales, e.g. the coastal current around Antarctica that proves to play a major role in the sea ice dynamics. It is worth noting that for ocean and sea ice configurations forced by atmospheric reanalysis the eddy-permitting resolution is commonly used (within and outside the NEMO community). Our model with a horizontal resolution of 1/4° is however far from the forefront of current high-resolution modelling. To employ a model with much lower resolution, as in climate studies, will not give satisfactory results for our purposes, mainly due to the weak representation of the Antarctic coastal current. Evaluating the impact of the model resolution on sea ice properties and its variability is beyond the purpose of this study. Anyway, our group has recently performed an ocean/sea ice simulation at eddying resolution (Iovino et al. 2016) that clearly shows the improvements of the ocean circulation mainly in continental shelf zones and coastal areas.

Although using a low-resolution grid reduces the computational cost and so gives the possibility to run longer simulations, we do strongly think that to employ a model with lower resolution, as in climate studies, will not give satisfactory results for our purposes, mainly due to the weak representation of the Antarctic coastal current. Evaluating the impact of the model resolution on sea ice properties and its variability is beyond the purpose of this study.

Iovino, D., Masina, S., Storto, A., Cipollone, A., and Stepanov, V. N.: A 1/16° eddying simulation of the global NEMO sea-ice–ocean system, Geosci. Model Dev., 9, 2665-2684, doi:10.5194/gmd-9-2665-2016, 2016.

Specific comments:
p.1, l.25: "in situ observations show a prevailing atmospheric and oceanic warming trend", is that so? The situation is not that simple for Antarctica and the Southern Ocean. See Jones et al. 2016. If this is an important point a better discussion and citations are necessary otherwise I wouldn't make this statement.
Jones, J. M., Gille, S. T., Goosse, H., Abram, N. J., Canziani, P. O., Charman, D. J., … Vance, T. R. (2016). Assessing recent trends in high-latitude Southern Hemisphere surface climate. Nature Climate Change, 6(10), 917–926. http://doi.org/10.1038/nclimate3103
We thank the Referee for making us aware of the study by Jones et al. (2016).
We do agree of course that, as also indicated in the paper, the climate variability in this area is very complex, and that climate change does not have a uniform impact over the entire Antarctic region. Observations confirm that the surface air temperature features – predominantly – a warming trend over the 1979-2014 period. On the contrary, the overall sea surface temperature (SST) exhibits a cooling trend (with the exception of the southeast Indian Ocean sector, and in the Weddell, Bellingshausen and Amundsen Seas), which is associated with the observed expansion of Antarctic sea ice. Whereas in most regions the upper layers (~100 m) of the Antarctic Ocean have cooled during the past decades, the remainder of the upper 1000m warmed significantly since the 1930 in conjunction with climate warming (e.g. Gille, 2008, Yin et al. 2011).
To avoid misunderstanding, we have clearly specified that we do not refer only to the surface temperature. We modified the sentence as follows: '*Even though satellite observations show a prevailing warming of the surface air (Jones et al., 2016) and the upper-ocean (Gille 2008), they also indicate that Antarctic sea ice has been expanding over the past few decades (e.g. Vaughan et al., 2013).*' (p. 1, l. 23-25)

Gille, S. T. Decadal-scale temperature trends in the Southern Hemisphere ocean. J. Clim. 21, 4749-4765, 2008.
Yin, J., Overpeck, J.T., Griffies, S.M., Hu, A., Russell, J.L. and Stouffer R.J.: Different magnitudes of projected subsurface ocean warming around Greenland and Antarctica. Nature Geosci., 4(8), 524–528 , 2011.

p.1, l.26: The reference to Collins et al. 2013 seems strange. This IPCC chapter is not about observation of the last decades but about projection with climate models. I think chapter 4 would be better suited here.
The reviewer is right, we apologize for the mix-up. We changed the reference to the IPCC chapter on the observations in the cryosphere referring to Vaughan et al., 2013.

Vaughan, D. G., Comiso, J. C., Allison, I., Carrasco, J., Kaser, G., Kwok, R., Mote, P., Murray, T., Paul,

F., Ren, J., Rignot, E., Solomina, O., Steffen, K. and Zhang, T.: Observations: Cryosphere, *in:* Climate Change 2013: The Physical Science Basis*, Contribution of Working Group I to the Fifth Assessment Report of the Intergovernmental Panel on Climate Change, edited by* T. F. Stocker *et al., Cambridge Univ. Press,* Cambridge, UK, 2013.

p.2, l.6-7: This is too vague. You need to explain this feedback a little more. Also note that the feedback described by Goose and Zunz 2014 is only amplifying an initial anomaly that could come from wind, rain, Antarctic melt water.

Any feedback is only amplifying an initial anomaly. That the atmosphere is variable and changes from year to year and causes disturbances at the surface of the ocean is also well known. In our opinion, the fact that such a disturbance is able to trigger a process that is sustained and can cause a multi-decadal trend in the sea ice is more important than the nature of the initial disturbance. To shed more light on the feedback mechanisms, we expanded the paragraph:

*'Zhang (2007) suggests that the trends in the Antarctic sea ice extent can be explained as the result of a feedback between the sea ice and the upper ocean stratification. He attributes the increase in sea ice to a reduced capability of the ocean to melt sea ice. Enhanced surface air temperatures and longwave radiation in his model lead to increased stratification and reduced vertical heat flux in the ocean. Similarly, Goosse and Zunz (2014) argue that the multidecadal variability of the Southern Ocean can explain the recent trends in Antarctic sea ice in connection with small changes of the pycnocline that increase the ocean's stratification. Also Bintanja et al. (2013) attributes the change in sea ice to a fresher surface layer, but sees the cause in the enhanced influx of meltwater from the Antarctic ice sheet.'* (p. 2, l. 7-13)

p.12, l.30: Should be a reference to Figure 4? Figure 4 and 5 are referenced incorrectly in the rest of the text.

Yes. We apologize for the error. We corrected the references to Figures 4 and 5:
*'… sea ice starts to decrease (Figure 4). '* (p. 12, l. 6)
*'The seasonal mean (Figure 4) still bears …'* (p. 12, l. 8)

[revised manuscript text omitted]